# A Hybrid Parallel Balanced Phasmatodea Population Evolution Algorithm and Its Application in Workshop Material Scheduling

**DOI:** 10.3390/e25060848

**Published:** 2023-05-25

**Authors:** Song Han, Shanshan Chen, Fengting Yan, Jengshyang Pan, Yunxiang Zhu

**Affiliations:** 1School of Electronic and Electric Engineering, Shanghai University of Engineering Science, Shanghai 201620, China; 2College of Computer Science and Engineering, Shandong University of Science and Technology, Qingdao 266590, China

**Keywords:** phasmatodea population evolution algorithm, equilibrium optimization algorithm, hybrid method, grouping and parallelism, workshop material scheduling

## Abstract

The phasmatodea population evolution algorithm (PPE) is a recently proposed meta-heuristic algorithm based on the evolutionary characteristics of the stick insect population. The algorithm simulates the features of convergent evolution, population competition, and population growth in the evolution process of the stick insect population in nature and realizes the above process through the population competition and growth model. Since the algorithm has a slow convergence speed and falls easily into local optimality, in this paper, it is mixed with the equilibrium optimization algorithm to make it easier to avoid the local optimum. Based on the hybrid algorithm, the population is grouped and processed in parallel to accelerate the algorithm’s convergence speed and achieve better convergence accuracy. On this basis, we propose the hybrid parallel balanced phasmatodea population evolution algorithm (HP_PPE), and this algorithm is compared and tested on the CEC2017, a novel benchmark function suite. The results show that the performance of HP_PPE is better than that of similar algorithms. Finally, this paper applies HP_PPE to solve the AGV workshop material scheduling problem. Experimental results show that HP_PPE can achieve better scheduling results than other algorithms.

## 1. Introduction

Modern heuristic algorithms and meta-heuristic algorithms are other names for intelligent optimization algorithms. They are algorithms built on intuition or empirical data. They can find approximately optimal solutions to complex problems in a limited time. They excel at solving nonlinear, global, combinatorial, and other problems. Manufacturing scheduling, path planning, signal and graphics processing, wireless sensor networks, automatic control, and other fields can all be used successfully. As a result, research into intelligent optimization algorithms is critical.

Many problems are insurmountable in a reasonable amount of time. A meta-heuristic algorithm is typically used to obtain an approximately optimal solution to such problems, and its principles are mostly derived from existing phenomena in the real world [1]. It is widely used to solve complex optimization problems, such as those in industry and economics. Meta-heuristic algorithms are extensions of heuristic algorithms, which are broadly classified as swarm intelligence algorithms, evolutionary algorithms, and physical algorithms. Here are some typical examples of swarm intelligence algorithms: cuckoo search algorithm (CSA) [2,3], artificial bee colony algorithm (ABC) [4,5], bat algorithm (BA) [6], salp swarm algorithm (SSA) [7], pigeon optimization algorithm (PIO) [8], etc. They are global optimization algorithms with continuous variables and multiple objectives. Evolution-based metaheuristic algorithms include the differential evolution algorithm (DE) [9,10], symbiotic search algorithm (SOS) [11], quasi-affine transformation evolution algorithm (quatre) [12], the genetic algorithm (GA) [13,14], etc. Finally, meta-heuristic algorithms based on physical models include the simulated annealing algorithm (SA) [15,16], gravity search algorithm (GSA) [17], charged system search algorithm (CSS) [18], etc.

Faramarzia proposed the equilibrium optimization algorithm (EO) [19] in 2019, a physics-based meta-heuristic algorithm. The principle of the algorithm is to control the mass-volume balance model using a random exploration mechanism. Each particle in EO updates its position by randomly selecting a particle from the equilibrium candidate solution and finally reaching equilibrium. The adaptive value of the algorithm’s control parameters can reduce the particle’s moving speed. The ability to search and develop is largely determined by this random update strategy and the control parameters in the algorithm [20].

The phasmatodea population evolution algorithm (PPE) was proposed by Song in 2020. It is a meta-heuristic algorithm based on swarm intelligence and evolution. PPE considers the solution as a population of stick insects, which has the property of population size in addition to the corresponding fitness values. Among them, the trend of population evolution is affected by the size of its population. A population will make different decisions according to its size, which is also affected by environmental changes to a certain extent [21].

As a new algorithm, PPE is theoretically simple and easy to implement. However, its convergence speed and global search capability need to be improved further to stay within the local optimum. Based on the concept of hybrid and parallel strategy, this paper combined the phasmatodea population evolution algorithm with the equalization optimization algorithm to improve the algorithm’s diversity, expand the algorithm’s search range, and solve the problems that the algorithm fell easily into local optimum, had slow convergence speed, and had low convergence accuracy.

There are many strategies to improve the performance of meta-heuristic algorithms, such as chaotic mapping, Levi’s flight, parallel strategy, the Gaussian walk strategy, the random walk strategy, the sine and cosine optimization strategy, the adaptive strategy, the hybrid algorithm improvement strategy, etc. Among them, the packet parallel and hybrid strategies are two important algorithm improvement strategies. The discussion about parallel and hybrid optimization algorithms is multifaceted, and different algorithms are applied in different fields. The packet parallel strategy expands the original population into multiple groups and performs inter-group communication in the iteration process to improve the convergence speed of the algorithm [22]. For example, parallel particle swarm optimization (PSO) [23,24,25], parallel ant colony optimization (ACO) [26], the parallel equilibrium optimizer algorithm (PEO) [27], etc. Different inter-group communication strategies greatly impact the algorithm’s performance, and there are many studies on inter-group communication strategies. In order to solve the inherent defects of a single optimization method, many algorithms use hybrid improvement methods to fuse algorithms with different characteristics and make use of the positive features of the sub-algorithms to make the performance of the hybrid algorithm better than the sub-algorithms. For example, the hybrid Van der Waals force bee colony algorithm (VPACO) [28] was used to solve the classical TSP problem, the hybrid Harris Hawks Optimization algorithm (hHHO-AOA) [29] was used to optimize the size and design of autonomous microgrids, and the hybrid Genetic and Particle Swarm Optimization algorithm (GA-PSO) [30] optimized the parameters of the five-parameter model. In this paper, the two methods are combined; the balanced optimization algorithm is mixed with the phasmatodea population evolution algorithm through the hybrid method, and the packet parallel strategy is applied to the hybrid balanced phasmatodea population evolution algorithm. On this basis, a new hybrid parallel balanced phasmatodea population evolution algorithm called HP_PPE is proposed. The algorithm’s performance is compared and tested by the CEC2017 test suite, which is relatively new worldwide.

In order to test the ability of HP_PPE to solve practical problems, we take the AGV workshop material scheduling problem as an application example of the algorithm. AGV is an automated guided vehicle. It is a flexible and efficient automated small transport vehicle widely used in the manufacturing field, storage systems, and other scenarios for cargo handling [31]. An AGV has the apparent advantages of good flexibility, effortless control, and high intelligence, and the transportation efficiency of the system can be conveniently improved by using AGVs for material transportation. Compared with the traditional manual or semi-manual mode, the transportation mode can reduce labor intensity, reduce the danger in the cargo handling process, and improve production efficiency [32]. In recent years, the AGV workshop scheduling problem has attracted the attention of a large number of scholars. At present, the main solving algorithms include the genetic algorithm (GA), particle swarm optimization (PSO), differential evolution algorithm (DE), Tabu search algorithm (TS), etc., as well as related improved algorithms. The load weight of each vehicle cannot exceed its maximum load, which is an essential constraint of AGVs. On this basis, the transportation time and path can be minimized to meet the time and efficiency requirements of the workshop’s production process.

The contributions of this paper are as follows:In this paper, we combined the hybrid method and grouped-parallel strategy and apply both of them to the study of PPE for the first time, and on this basis, we proposed the hybrid parallel balanced phasmatodea population evolution algorithm (HP_PPE), which significantly improves the optimization ability of the original phasmatodea population evolution algorithm.Secondly, the newly proposed algorithm is applied to the AGV workshop material schedule for the first time, which expands the application scenario of HP_PPE in the workshop production scenario.

The remainder of this article is organized as follows: Section 2 briefly introduces some preliminary knowledge. Section 3 formally presents our hybrid parallel phasmatodea population evolution algorithm in detail. Section 4 uses extensive experiments to evaluate the algorithm. Section 5 shows the further application analysis of the proposed HP_PPE algorithm in AGV workshop material scheduling. Section 6 presents conclusions about the existing work.

## 2. Related Work

The research in this paper is a study of a novel metaheuristic algorithm known as the phasmatodea population evolution algorithm; then, we propose our own algorithm based on this novel algorithm, and finally apply our newly proposed algorithm to solve the workshop material scheduling problem. In the following, we briefly introduce some of the already existing theoretical foundations of this research.

### 2.1. Phasmatodea Population Evolution Algorithm (PPE)

PPE primarily simulates some characteristics in the evolution process of stick insect populations in nature, such as convergent evolution, path dependence, population mutation, population growth, and population competition, and develops the algorithm based on the characteristics above [21]. We briefly describe the first two most important features, which are convergent evolution and path dependence. Convergent evolution refers to the possibility of similar evolution occurring if multiple populations live in similar environments. Path dependence implies that a population’s evolutionary trend will change as its living conditions change. Unlike other algorithms that take individuals as the solution, PPE takes the stick insect population as the solution, and its population has the fitness value and also the property of population size. In the process of seeking the optimal solution, the convergent evolutionary mechanism makes the size of the population change continuously, and path dependence affects the evolutionary trend. The population position change is controlled by the evolutionary trend, and the new position of the population is the sum of the old position and the evolutionary trend. After continuously iteratively updating the properties of the current population, the algorithm finds the optimal solution.

PPE is mostly implemented using the population growth and population competition models. Equation (1) is commonly used to represent the population growth model, and the specific implementation employs a logical difference Equation (2). The Equation (3) describes the population competition model.
(1)$$\frac{dp}{dt}=rp\left(1\;-\;\frac{p}{K}\right)$$
(2)$$p^{t+1}{=ap}^{t}{(1-p}^{t})$$
(3)$$\frac{dp}{dt}{=r}_{1}p(1-\frac{p}{n_{1}}-s_{1}\frac{q}{n_{2}})$$

In Equation (1), p is the population number, r is the population’s effective growth rate, and K is the population’s maximum environmental bearing capacity in space. In Equation (2), taking the value of *K* in Equation (1) as one, a represents the growth rate, p ranges from 0 to 1, and a ranges from 0 to 4. *t* represents the current number of iterations, *t* ranges from 0 to *Max_gen*, and *Max_gen* represents the maximum number of iterations. In Equation (3), *q* is the number of populations closest to the current population selected by the population competition condition, and $r_{1}$ is the population’s effective growth rate. In Equations (1) and (3), 1/*Np* is the initial number of current populations and *Np* is the total number of populations in the *N*-dimensional space. Population p has a maximum environmental carrying capacity of $n_{1}$, while population q has a maximum environmental carrying capacity of $n_{2}$. $s_{1}$ denotes that the number of units q consumes a multiple of the number of resources supporting p.

### 2.2. Equilibrium Optimization Algorithm (EO)

Each particle (solution) and its position (concentration) serve as search agents in EO. During the iteration phase, the search subject updates its current position (concentration) based on the equilibrium candidate at random until the completion of the iteration to acquire the best result (equilibrium state). In order to improve the search capability, an “equalization pool” is constructed in EO. Equilibrium candidates are the five particles in the equilibrium pool. The five particles are made up of the four particles’ best fitness values and the average of the first four [19].

The initial population consists of randomly distributed particles in the search space, as shown in Equation (4). The equilibrium pool vector is represented by Equation (5). Finally, the updated equation of EO is shown in Equation (6).
(4)$$C_{i}^{initial}=C_{i}^{min}+{rand}_{i}\left(C_{i}^{max}-C_{i}^{min}\right)\;i=1,2,3,\ldots n$$
(5)$${\vec{C}}_{eq,pool}=\{{\vec{C}}_{eq\left(1\right)},{\vec{C}}_{eq\left(2\right)},{\vec{C}}_{eq\left(3\right)},{\vec{C}}_{eq\left(4\right)},{\vec{C}}_{eq\left(ave\right)}\}$$

In Equation (4), $C_{i}^{initial}$ is the position (initial concentration) vector of the ith particle, $C_{i}^{min}$ is the minimum dimension of the ith particle, $C_{i}^{max}$ is the maximum dimension of the ith particle, $rand_{i}$ is a random vector of the ith particle and each element of the vector is in the interval [0,1], and *n* is the total number of particles. In Equation (6), $\vec{C}$ represents the particle to be updated. The vector $\vec{F}$ is used to maintain the balance between search and development, and the vector $\vec{G}$ is used to improve the accuracy of the solution in the development phase. λ is a random number in the interval [0,1]. *V* is the control volume. The first part, ${\vec{C}}_{eq}$ on the right-hand side of the equation, is the equilibrium candidate particles obtained in the equilibrium pool (${\vec{C}}_{eq,pool}$). The second and third terms represent a change in the concentration of particles in the population. The second term attempts to find an optimal solution in the solution space, and the third term helps to make the solution more accurate. During the iteration process, each particle is updated through an updated equation to improve the adaptability of the particle and the overall optimization ability of the algorithm [27].
(6)$$\vec{C}={\vec{C}}_{eq}+\left(\vec{C}-{\vec{C}}_{eq}\right)\cdot \vec{F}+\frac{\vec{G}}{\lambda V}\left(1-\vec{F}\right)$$

### 2.3. AGV Workshop Material Scheduling

AGV means automated guided vehicle. Considering that an AGV only engages in material transportation throughout the processing process, the material scheduling of the entire production shop is viewed as a small shop logistics system when addressing the problem of material distribution and scheduling in workshops. On this premise, a multi-AGV task scheduling model is suggested to enable AGV walking path optimization.

The architecture of the material area and production area, how the AGV moves, and the difficulty of material transportation all significantly impact the scheduling problem evaluation index in a workshop logistics system. The workshop material schedule is typically seen as a whole system to obtain a better scheduling effect. The essence of the job shop material distribution and scheduling problem is to minimize the overall trip distance of the AGV while assuring punctuality and economy, which is an extension of the limited route planning problem. According to the specific workshop processing process, under the constraints of the given material loading and unloading stations, processing stations, workshop layout and trolley walking rules, as well as the number of materials required for each processing step, various processing materials are transported by AGV to the target stations to meet the production requirements, and the best AGV trolley walking path to complete the above workshop material distribution scheduling requirements is solved. The AGV starts from the docking station inside a production beat, continuously conducts batching activities in a reciprocating cycle between the material warehouse and the station, and returns the finished goods to the warehouse for storage after each processing [33].

The objective function of the AGV workshop material scheduling model we denoted by *D*. The variables are *x* and *y*, as shown in Equation (7), and the constraints are shown in Equation (8).
(7)$$minD=\overset{n}{\underset{i=0}{\sum }}\overset{n}{\underset{j=0}{\sum }}\overset{k}{\underset{v=0}{\sum }}C_{ij}x_{ijv}$$
(8)$$\overset{n}{\underset{i=0}{\sum }}g_{i}y_{iv}\leq Q,v=1,2,\ldots ,k$$

In Equations (7) and (8), the parameter *v* represents the vehicle number in this paper, and the parameter *k* represents the total number of vehicles. $C_{ij}$ means the distance from material point *i* to material point *j*, *Q* represents the maximum load of the trolley, *n* is the total number of production points, and *g* represents the demand of each production point. The optimization objective of the AGV material scheduling model considered in this paper is to achieve the shortest scheduling distance for a certain load per AGV. For further description of this model, see Section 5.

## 3. Hybrid Parallel Balancing Phasmatodea Algorithm

We propose HP_PPE to solve the shortcomings of PPE, such as easy local optimization, low convergence accuracy, sluggish convergence speed, and lengthy time requirements. HP_PPE incorporates the equalization pool in the equalization optimization method into PPE using the hybrid improvement strategy. Based on the hybrid algorithm, it uses the parallel packet approach.

### 3.1. Hybrid Improvement Strategy

In order to solve the defects that the phasmatodea population evolution algorithm has shown, easily falling into localization and demonstrating low convergence accuracy, this paper adopts a hybrid improvement mechanism to mix the phasmatodea population evolution algorithm with other algorithms to improve the diversity and convergence accuracy of the original algorithm. The equilibrium optimization algorithm is well suited to be combined with different algorithms due to its fast convergence and simple structure. In this paper, we find that compared with other algorithms, the phasmatodea population evolution algorithm’s performance is best when mixed with the balanced optimization algorithm, so this paper combines the equilibrium optimization algorithm with the phasmatodea population evolution algorithm and adds an equilibrium pool to the phasmatodea population evolution algorithm. It records local optimal solutions, population fluctuations, and other activities. In the experimental section of this study, it is discovered that when the equilibrium optimization method is integrated into the phasmatodea population evolution algorithm, its capacity to jump out of the local optimum is considerably increased.

### 3.2. Parallel Communication Strategy

Many intelligent optimization algorithms include parallel improvement strategies, and by dividing the initial population, the parallel method can boost the algorithm’s global search ability. The grouping structure, movement strategy of each group, population renewal strategy, competition model, and establishment of the equalized pool in the hybrid parallel equalized phasmatodea population evolution algorithm (HP_PPE) are consistent with the original equalization algorithm, and the parameters of the equalized pool are updated to the new parameters. Populations in various groups communicate with one another frequently in order to promote group collaboration and the algorithm’s convergence speed. HP_PPE employs two modes of communication. The first technique is used in the early stages of the algorithm. It leverages inter-group communication to deal with additional mutations in the poor population, which can boost the algorithm’s convergence speed. The second technique is used later in the algorithm to replace the lousy population in each group, which can alleviate the problem of the algorithm being stuck in a local optimum.

### 3.3. Implementation of Hybrid Parallel Improvement Strategy

This algorithm mainly includes initialization, construction of an equalization pool, and communication between groups.

#### 3.3.1. Initialization

HP_PPE is initialized first. In the initialization process, HP_PPE takes the solution as the population of stick insects, and first initializes $N_{p}$ populations randomly; each population is represented by a point $x_{i}$ in the n-dimensional space, and each point is randomly generated under the constraints of the upper and lower boundaries. Each population $x_{i}$ has two attributes: the population number *p_i_* and the population growth rate *a_i_*. The initial population number $p_{i}$ of each population *x_i_* is calculated by Equation (9), and the initial value of each population growth rate *a* was set to 1.1. HP_PPE uses *k* historical optimal solutions to guide the movement of surrounding solutions. The formula for *k* is shown in (10), and all historical optimal solutions are stored in $H_{0}$. $H_{0}$ = [${xh}_{1}$,…,${xh}_{i}$,…,${xh}_{k}$].
(9)$$p_{i}=\frac{1}{N_{p}}$$
(10)$$k=\lfloor log\left(N_{P}\right)\rfloor +1$$
(11)$$x^{t+1}{=x}^{t}+{ev}^{t}$$
(12)$${ev}^{t+1}=\left({1-p}^{t+1}\right){A+p}^{t+1}{(ev}^{t}+m)$$
(13)$$A=\left(s\left(H_{o}{,x}^{t}\right)-x^{t}\right)\cdot c$$
(14)$$p^{t+1}=a^{t+1}p^{t}(1-p^{t})$$
(15)$$a^{t+1}=a^{t}(1+\frac{f(x^{t})-f(x^{t+1})}{f(x^{t+1})})$$

The population position is then updated. The position of the current population in the future is defined by its current position and the population evolution tendency. Equation (11) depicts the position update formula. The fitness value of the present population and the optimal global solution are calculated once the current population shifts to a new position. In Equation (11), $x^{t+1}$ is the position of the population at time t + 1, $x^{t}$ is the position of the population at time t, ev is the evolutionary trend of the population, and the update formula of ev is shown in Equation (12). In Equation (12), *A* represents the level closest to the nearest optimal, as shown in Equation (13). $s\left(H_{o},x^{t}\right)$ is used to find the closest historical optimal solution to $x^{t}$ in $H_{o}$, and *c* is set to 0.2. The updating procedure has three parts. The first and second parts mainly use the characteristics of the convergent evolution of the stick insect population, and the third part adopts the population competition model. The population number is updated as shown in Equation (14), and the population growth rate a is updated as shown in Equation (15).

The first and second parts of the update employ the values calculated from the population position to choose which update method to implement, where the population mutation will affect the population’s evolutionary trend. If a better value for the evolved population’s location is computed, the next update will continue the prior evolutionary trend. If the population position does not improve, the population will no longer follow the original trend and instead choose the nearest optimal solution, resulting in unanticipated perturbations. On this premise, the population trend updating formula in the second section is updated into Equation (16).
(16)$${ev}^{t+1}=rand\cdot A+st\cdot B$$

In Equation (13), A represents the degree closest to the nearest optimal solution, *s*(*H_o_*, *x^t^*) is used to represent the historical optimal solution closest to *x^t^* in *H_o_*, *c* is the influence coefficient of the nearest optimal solution on the population, and m represents the mutation of the population in some dimensions. The population size is updated as shown in equation (14), where the population growth rate *a* is updated as shown in equation (15). For Equation (16), rand denotes an n-dimensional random vector generated using uniform distribution, with each dimension between 0 and 1, and B denotes an n-dimensional random vector generated using standard normal distribution. st is initially set to 0.1 (*C_max_* − *C_min_*), and with each iteration of the algorithm, st is updated to st = 0.99st.

The third part is the influence of the competition between populations on the population evolution trend. First, the distance between two populations $x_{i}$ and $x_{j}$ is judged and compared with G. If the distance between two populations is less than G, the two populations will compete. The competition will have an impact on the evolution trend of the current population $x_{i}$. The calculation formula of G is shown in (15), the evolution trend is updated in Formula (18), and the population number $p_{i}$ in the competition mechanism is updated in formula (19). For Equation (17), Max_gen represents the number of iterations, and *t* represents the current number of iterations. For Equation (18), *x_i_* is the current population and $x_{j}$ is randomly selected from other $N_{p}$−1 populations.
(17)$$G=0.1\times \left(C_{max}-C_{min}\right)\frac{\text{Max\_gen}+1-t}{\text{Max\_gen}}$$
(18)$${ev}^{t+1}{=ev}^{t+1}+\frac{f\left(x_{j}\right)-f\left(x_{i}\right)}{f\left(x_{j}\right)}(x_{j}-x_{i})$$
(19)$$p_{i}{=p}_{i}{+a}_{i}p_{i}(1-p_{i}-\frac{f\left(x_{j}\right)}{f\left(x_{i}\right)}p_{j})$$

#### 3.3.2. Construction of a Balanced Pool

For the parallel grouping strategy, different grouping has a great impact on the performance of the algorithm. In this paper, we found that the performance of HP_PPE was best when the number of groups was 2, better than that of HP_PPE when the number of groups was zero, or more than two groups. Therefore, the number of groups of HP_PPE was set to 2. The fitness values of all populations are calculated and sorted in each group to obtain the optimal solution and four equilibrium candidate points. By comparing all groups, we can obtain the globally optimal solution *BestX* of the whole population and four global optimal equilibrium candidate points *C_eq_*_1_, *C_eq_*_2_, *C_eq_*_3_ and *C_eq_*_4_, so as to construct the equilibrium pool.

#### 3.3.3. Inter-Group Communication

Each group evolves individually when the equalization pool is initiated and constructed. After a specified number of iterations, the two policies are performed to communicate across groups, and the number of communication iteration intervals is set to 20. The first technique is employed in the algorithm’s early iterations to increase the algorithm’s convergence speed by dealing with specific mutations in the lousy population. The second technique is employed later in the algorithm iteration process to improve the algorithm’s development ability and search accuracy by replacing the lousy population in each group.

The initial inter-group communication technique is implemented in the first third of the total number of iterations. This approach individually mutates the particles with low fitness in each group, and the mutation equation is presented in (20). This strategy can accelerate the mutant particles’ approach to the average value of the equilibrium candidate solution and the ideal solution, increasing the algorithm’s convergence speed. $X_{d}$ and *BestX_d_* are the particles to be mutated and the global optimal particle, and $Ceq_{d}$ is the equilibrium candidate particle. rand is a random value between 0 and 1. The value of $a_{2}$ is determined by Equation (21).
(20)$$X_{d}=\frac{{\mathrm{BestX}}_{d}+{\mathrm{Ceq}}_{d}}{2}(a_{2}+\mathrm{rand}a_{2})$$
(21)$$a_{2}=\frac{\mathrm{Iter}}{\text{Max\_iter}}$$

The second inter-group communication technique is used in the last two-thirds of the iteration. It substitutes certain particles with low fitness with the average value of the ideal particles in each group. The mean of the optimal particles in the first and second groups replaces the unfit particles in the first group. This process is repeated until some less-fit particles in all groups have been replaced.

Finally, after the parameters are modified and the optimal value is obtained, the iteration cycle is complete, and a new iteration begins, which will continue until the process is complete. Algorithm 1 and Figure 1 show the HP_PPE flowchart and pseudo code, respectively.
**Algorithm 1:** *HP_PPE**1.  Initialize Np populations;**2.  Initialize ev, p, k and a;**3.  Group and initialize the position of the each group of populations randomly;**4.  Calculate fitness f (x), set the global optimal solution gbest and Ho;**5.  for t = 2 to Maxgen do**6.   for g = 1 to groups do**7.  for i = 1 to num_pop/groups do**8.    Update each x to newx; **9.    Calculate new fitness f (newx), **10.   update gbest and Ho; **11.   Update ai and pi; **12.   if f (newx) ≤ f (x) then **13.    update x, x = newx, update f (x); **14.    Update evi; **15.   if f (newx) > f (x) then **16.    if rd < pi then**17.     Update x, x = newx, update f(x); **18.    Update evi; **19.   Randomly choose a solution xj, (j ≠ i); **20.   if dist(xj, xi) < G then**21.     Update pi, update evi;**22.   if pi ≤ 0 or ai ≤ 0 or ai > 4 then **23.    Eliminate xi and replace it;**24.   find the equilibrium candidate populations of each group;**25. end for**26.   end for**27.   for g = 1 to groups do**28.  Compare, find the equilibrium candidate populations of global**29.  and global optional value;**30.   end for**31.   Calculate the a2**32.   if rem(iter,20) = 0 then**33.  for g = 1 to groups do**34.    each group is sorted according to the fitness value;**35.    if iter <= 1/3Max_iter**36.    use communication strategy one for half of the **37.    populations in each group;**38.    else iter > 1/3Max_iter**39.    use communication strategy two for half of the **40.    populations in each groups;**41.    end if**42.  end for**43.  for g = 1 to groups do**44.    for i = 1 to num_pop/groups do**45.   Calculate the fitness value for each population after **46.   the communication;**47.   find the equilibrium candidate populations of each group**48.    after the communication;**49.    end for**50.  end for**51.  for g = 1 to groups do**52.    Compare, find the equilibrium candidate populations of **53.    global after the communication;**54.  end for**55.   end if**56.   The optimal comparison between each population in each group,**57.   and the individual population so far was conducted to select **58.   the population with good fitness;**59.   for g = 1 to groups do**60.  Structural Equilibrium pool.**61.  for i = 1 to num_pop/groups do**62.    Update each population;**63.  end for**64.   end for**65. end for*

## 4. Experimental Analysis of HP_PPE

In this section, we test the performance of HP_PPE using the internationally novel CEC2017 test function suite and compare it with other algorithms. First, it compares HP_PPE with some standard algorithms, and then it compares HP_PPE with some parallel algorithms. At the end of each section, we analyze the comparison results.

### 4.1. Benchmark Functions

This section tests our proposed HP_PPE using 30 functions from the CEC2017 benchmark function suite, as shown in Table 1. The benchmark functions include three unimodal functions (F1–F3), seven simple multimodal functions (F4–F10), ten hybrid functions (F11–F20), and ten composition functions (F21–F30). The hybrid functions are composed of the first two functions. Among them, the unimodal functions test the development ability, the multimodal functions test the exploration ability, and the hybrid and composition functions are used to represent some challenging problems. All test functions are minimization problems, and the search range of all test functions is set to [−100, 100]. Since most variables in real-world problems have few connections, the variables in CEC2017 are randomly divided into sub-components [34].

### 4.2. Comparison with Other Standard Algorithms

In this section, comparative experiments are conducted between HP_PPE and PPE [21], CCS [35], EBH [36], BH [37], WOA [38], and APSO [39], demonstrating the performance of the HP_PPE algorithm in low and high dimensions, as shown in Table 2 and Table 3.

General parameter settings for the experiments:For the two comparative experiments in this section, the evaluation time of all algorithms is set as 20,000 times.The population size of all algorithms is set as 20.The dimension in Table 1 is set as 10.The dimension in Table 2 is set as 30.The independent running times of each algorithm on different functions is set as 30.

The search range for each dimension is [−100, 100]. FES = 20,000, pop_size = 20, Dim_1 = 10, Dim_2 = 30, runs = 30.

Table 2 and Table 3 show the average values of the proposed HP_PPE and other comparison algorithms over 30 benchmark functions. In the last row of the table, the comparison results of all the functions are summarized. The symbol (<) means that the algorithm performs worse than HP_PPE on the current benchmark function, the symbol (=) means that the two algorithms perform similarly on the current benchmark function, and the symbol (>) means that HP_PPE performs poorly. The values in bold in the table are the best results for the current test function.

We performed analysis of the data in Table 2, and longitudinally compared the evaluation results of APSO, WOA, BH, EBH, CCS, and PPE on 30 benchmark functions, which were 22/0/8, 30/0/0, 28/0/2, 25/0/5, 23/0/7, and 22/0/8, respectively. This shows that HP_PPE has high convergence accuracy in 10 dimensions, and the overall performance is substantially ahead of the other six algorithms, especially WOA and BH. Comparing the experimental results of different reference functions horizontally shows that HP_PPE can achieve satisfactory results on most of the benchmark functions. Among them, HP_PPE can obtain the actual minimum value on F6-F12, F17-F18, F20, F23, F25-F26, and F29-F30. Compared with other algorithms, HP_PPE has the best ability to solve most multimodal functions, hybrid functions, and combinations, but is inferior to some algorithms in solving unimodal functions. Compared with APSO, HP_PPE is superior in solving most multimodal functions, most hybrid functions, and all combinatorial functions, although HP_PPE is inferior to APSO in solving unimodal functions. Compared with WOA, HP_PPE achieves better results on all functions. Compared with BH, HP_PPE has better performance on all unimodal functions, multimodal functions, hybrid functions, and most combinatorial functions. Compared with EBH, the ability of HP_PPE to solve unimodal functions is inferior. However, the ability of HP_PPE to solve all multimodal functions is superior to EBH, and the ability of HP_PPE to solve most diverse functions and most combined functions is superior to EBH. HP_PPE is superior to CCS in solving most unimodal functions, all multimodal functions, most diverse functions, and most combined functions. Compared with PPE, the ability of HP_PPE to solve unimodal functions was inferior, but the ability of HP_PPE to solve the other three types of functions was superior to PPE. Thus, in the case of 10 dimensions, HP_PPE is far ahead of other algorithms in its ability to solve multimodal, hybrid, and combined functions. However, its ability to solve unimodal functions is not as good as some algorithms.

We performed analysis of the data in Table 3; the evaluation results of APSO, WOA, BH, EBH, CCS, and PPE on 30 benchmark functions were 15/0/15, 26/0/4, 30/0/0, 21/0/9, 29/0/1, and 17/0/13, respectively, when compared longitudinally. It can be seen that the overall performance of HP_PPE is similar to that of the APSO algorithm, while it is significantly ahead of WOA, BH, CCS, EBH, and PPE. Comparing the best experimental results of different benchmark functions horizontally, it can be seen that HP_PPE is overall ahead of the other algorithms except APSO. Among them, HP_PPE can obtain the actual minimum value on F5, F8, F10, F15–F17, F19–F20, F26, and F29. Compared with APSO, although HP_PPE is not as good at solving unimodal functions, it has better abilities in solving multimodal functions and combinatorial functions than APSO, and its abilities in solving hybrid functions are similar to APSO. Compared with WOA, HP_PPE is superior in solving all unimodal functions, multimodal functions, hybrid functions, and most combinatorial functions. Compared with BH, HP_PPE achieves better results in solving all types of functions. Compared with EBH, HP_PPE is less capable of solving unimodal functions than EBH. However, the ability of HP_PPE to solve most multimodal functions and most hybrid functions is ahead of EBH. The ability of HP_PPE to solve combinatorial functions is similar to that of EBH. Compared with CCS, the ability of HP_PPE to solve all unimodal functions, all multimodal functions, and all mixed functions is superior, and the ability to solve most combined functions is superior. The ability of HP_PPE to solve unimodal functions is inferior to PPE, but the ability of HP_PPE to solve most multimodal functions, most mixed functions, and most combined functions is superior to PPE, which is consistent with the comparison results of 10 dimensions. Thus, in the case of 30 dimensions, the ability of HP_PPE to solve unimodal functions is not as good as that of partial algorithms. The ability to solve multimodal functions is far ahead of other algorithms, which is consistent with the 10-dimensional results. The ability to solve the mixed functions, although approximate to APSO, is better than other algorithms. The ability to solve combinatorial functions, although approximate to EBH, is ahead of other comparison algorithms.

In order to further compare and analyze the performance of HP_PPE, Table 4 and Table 5 show the best results obtained by the four test functions in the cases of 10 and 30 dimensions, respectively. As can be seen from Table 4, compared with other algorithms, the overall performance of HP_PPE is better than the other six classical algorithms. In particular, it is far ahead of other algorithms in terms of multimodal functions, mixture functions, and combination functions, which is consistent with the conclusions obtained from the above analysis. As can be seen from Table 5, the overall performance of HP_PPE is better than five other algorithms, although it is second to APSO. Unlike in ten dimensions, although HP_PPE’s ability to solve hybrid functions is ahead of other algorithms, its ability to solve multimodal functions is approximated by APSO, and its ability to solve combinatorial functions is slightly inferior to APSO and approximated by PPE and EBH.

Although the above comparison results clarify the optimization performance of HP_PPE, the Wilcoxon signed rank test is also used in this paper to verify the statistically significant difference between HP_PPE and other algorithms. In this statistical test, the null hypothesis indicates that the median of the results obtained from HP_PPE is not statistically significantly different from the other algorithms. In order to reject the null hypothesis, R+ and R− were calculated according to the results in Table 2 and Table 3, and the Wilcoxon signed rank test results of HP_PPE and the other six algorithms at the significance level of α = 0.05 in the case of 10 and 30 dimensions are shown in Table 6 and Table 7, respectively. “R+” in the table indicates the sum of the rankings of the test functions where HP_PPE exceeds the competitors, while “R−” is the reverse. The p-value records the significance of HP_PPE compared with the other algorithms. These values are evaluated to determine whether to accept or reject the null hypothesis [40]. “Sig.” indicates the significance result of HP_PPE compared with other algorithms; the symbol (−) indicates that the performance of the algorithm is better than that of HP_PPE under the current benchmark function, the symbol (+) means that the algorithm is worse than HP_PPE, and the symbol (≈) indicates that the performance of the algorithm is almost no different from that of HP_PPE. The comparative results of Table 6 and Table 7 are similar. The Wilcoxon signed rank test shows that the null hypothesis is rejected, which means that the optimization performance of HP_PPE is statistically far better than that of its competitors. From the perspective of statistics, it can be concluded that the optimization results of HP_PPE are significantly better than those of WOA, BH, EBH, and CCS, and there is no significant difference between HP_PPE and APSO and PPE.

### 4.3. Convergence Analysis

Meta-heuristic algorithms with different convergence speeds may eventually obtain similar results, so the analysis of algorithm convergence is also a critical experimental link. In this section, fourteen groups are selected from the 10-dimensional experimental convergence diagram for demonstration, including one unimodal function (F2), five multimodal functions (F4, F6, F7, F8, and F9), three hybrid functions (F13, F14, and F17), and five combined functions (F21, F22, F26, F28, and F30). Figure 2 shows the convergence curves of the seven algorithms under different test functions. The horizontal axis is the number of iterations, and the vertical axis is the logarithmic scale of the convergence value of the current function.

Experimental results show that the convergence accuracy of HP_PPE is better than that of other algorithms. HP_PPE can achieve continuous convergence on most benchmark functions, and its all-around performance is better than other algorithms. In the early stages of algorithm execution, HP_PPE has a faster convergence speed than other algorithms on most benchmark functions. Compared with other algorithms, the proposed HP_PPE algorithm does not easily fall into the local optimal state. As shown in Figure 2a, although the convergence speed of HP_PPE was slower than that of APSO in the early stage, it achieved convergence results similar to APSO in the later stage. As shown in Figure 2h, although the convergence speed of HP_PPE was not as good as that of APSO, it achieved a relatively good convergence result in the later stages. As illustrated in Figure 2c, HP_PPE converged more stably than other algorithms without entering a local optimum, resulting in better convergence results. As shown in Figure 2d, although the convergence speed was not as fast as other algorithms in the early stage, other algorithms fell into local optima, while HP_PPE stably converged to a better value. As shown in Figure 2f, APSO, EBH, and PPE all fell into local optima, while HP_PPE converged more stably and achieved better convergence results than BH, CCS, and WOA. As shown in Figure 2h, although the convergence speed and final convergence value of HP_PPE were not as good as those of APSO, compared with the other five algorithms, HP_PPE had a faster convergence speed and higher convergence accuracy. By analyzing other convergence graphs, it can be found that the convergence speed of HP_PPE is faster than other algorithms, it is jumps out of the local optimum more easily, and the convergence accuracy is higher than that of the other algorithms.

Most of the comparison algorithms do not continue to converge, and the convergence curve tends to be stable in the later stages, which can be regarded as the algorithm falling into the local optimum. At the same time, HP_PPE can find better results and jump out of the local optimum. By comparing with PPE, it can be found that the HP_PPE algorithm improves the diversity of the algorithm through parallel mechanisms and hybrid utilization of the advantages of equilibrium optimization algorithms and PPE, and solves the defects of PPE, such as its slow convergence speed and tendency to fall into a local optimum.

### 4.4. Comparison with Parallel Algorithms

In order to further verify the performance of HP_PPE, three parallel algorithms, PPSO [41], PWOA [42], and MMSCA [43], are selected in this paper to compare with it. Table 8 and Table 9 show the average test values of HP_PPE and the other comparison algorithms on 30 benchmark functions in CEC2017 in 10 and 30 dimensions, respectively. The parameters and symbols of the algorithm are set as those in Section 4.2 of this paper. HP_PPE can obtain the best results in most functions, and the overall performance is better than other parallel algorithms. The data in Table 8 were analyzed, and the evaluation results of PPSO, PWOA, and MMSCA on 30 benchmark functions were compared vertically, with results of 21/0/9, 19/0/11, and 28/0/2, respectively. It can be seen that the overall performance of HP_PPE is ahead of the three parallel algorithms. Comparing the best experimental results of different benchmark functions horizontally shows that HP_PPE can achieve satisfactory results on most benchmark functions. Among them, HP_PPE can obtain the actual minimum value on F2, F5-F8, F10-F13, F15-F17, F19, F21, F25-F26, and F28-F30. Compared with the other three parallel algorithms, HP_PPE has the best solving capability for most multimodal, hybrid, and combinatorial functions, although it is inferior to some of the algorithms in solving unimodal functions. The data in Table 9 were analyzed, and the evaluation results of PPSO, PWOA, and MMSCA on 30 benchmark functions were compared longitudinally, with results of 25/0/5, 23/0/7, and 27/0/3, respectively. It can be seen that the overall performance of HP_PPE is ahead of the other three parallel algorithms. Comparing the best experimental results of different benchmark functions horizontally shows that HP_PPE can achieve satisfactory results on most benchmark functions. Among them, HP_PPE can obtain the actual minimum value on F2, F4-F11, F17-F18, F20, F22, and F25-F26. Compared with the other three parallel algorithms, HP_PPE has the best ability to solve most multimodal and combination functions, although it is inferior to some algorithms in solving unimodal and mixed functions.

In order to further compare the overall performance of HP_PPE with other parallel algorithms, the best results obtained by the four algorithms on the four test functions in the cases of 10 and 30 dimensions are shown in Table 10 and Table 11, respectively. As can be seen from Table 10, compared with other parallel algorithms, the overall performance of HP_PPE is better than the other three classical algorithms. In particular, its performance for multimodal and hybrid functions is ahead of other algorithms. Table 11 shows that, compared with other parallel algorithms, the overall performance of HP_PPE is better than the other three classical algorithms, consistent with the 10-dimensional comparison results. In particular, it is superior to the other three parallel algorithms on multimodal, hybrid, and combinatorial functions.

Table 12 and Table 13 show the Wilcoxon signed rank test results between HP_PPE and the other three parallel algorithms. By analyzing the data in Table 12, it can be found that in the case of 10 dimensions, the performance of HP_PPE is significantly better than that of PWOA and similar to that of PPSO and MMSCA. By analyzing the data in Table 13, it can be found that the performance of HP_PPE is significantly better than that of PPSO, MMSCA, and PWOA in the case of 30 dimensions. This can further illustrate the superiority of the HP_PPE algorithm.

Figure 3 shows the convergence plots of the four parallel algorithms under different test functions. In this section, a unimodal function (F2), five multimodal functions (F4, F7, F8, F9, and F10), six hybrid functions (F11, F12, F14, F17, F18, and F19), and six combinatorial functions (F22, F24, F25, F26, F28, and F29) are selected. Comparing convergence speeds, MMSCA and PWOA are significantly slower than HP_PPE. PPSO and HP_PPE are similar in the early stages, but the convergence rate of PPSO is slower in the later stages. From this, we can infer that HP_PPE is more likely to get rid of the local optimum and continue to seek the global minimum at a later stage. Therefore, the overall convergence performance of the algorithm is better than that of the other three parallel algorithms.

## 5. Applied to AGV Workshop Material Scheduling

We use HP_PPE to address this part’s AGV workshop material scheduling problem. The following is a general description of the workshop material scheduling problem: Multiple production points have various needs for products, and the material points supply the vehicles to transfer goods to the production points. When the truck departs from the material point, it delivers the items to the unallocated production location before returning to the material point. Unlike the usual AGV material schedule, the optimization goal of this work is to achieve the shortest scheduling distance for each AGV under a given load. Figure 4 depicts a two-dimensional shop scheduling environment with a single warehouse and numerous material points. The gray regions represent impediments, the green area represents the warehouse (the starting point for the AGV), and the blue areas represent the workstations.

### 5.1. Construction of AGV Workshop Material Scheduling Model

The following conditions are assumed to be true for this paper: (1) The material storehouse and each station are determined; (2) There is one material area and multiple production areas; (3) Each AGV can accept the current production order; (4) A material warehouse distributes materials to multiple stations within the production rhythm of a unit; (5) Each AGV’s carrying capacity and AGV running speed are limited to a specific range; (6) Each AGV’s initial batching position is determined randomly.

The parameter *v* (*v* = 1, 2,… *k*) represents the vehicle number in this paper, and the parameter *k* represents the total number of vehicles. The variables 0 and *i*(*i* = 1, 2, … n) were specified for the material and manufacturing areas. The following variables are defined:(22)$$x_{ijv}=\left\{\begin{array}{c} 1,\;\mathrm{if}\;\mathrm{AGV}\;v\;\mathrm{travels}\;\mathrm{from}\;\mathrm{station}\;i\;\mathrm{to}\;\mathrm{station}\;j,\\ 0,\;\mathrm{else} \end{array}\right.$$
(23)$$y_{iv}=\left\{\begin{array}{c} 1,\;\mathrm{if}\;\mathrm{station}\;i\;\mathrm{needs}\;\mathrm{to}\;\mathrm{be}\;\mathrm{fulfilled}\;\mathrm{by}\;\mathrm{AGV}\;v,\\ 0,\;\mathrm{else} \end{array}\right.$$

$C_{ij}$ means the distance from material point *i* to material point *j*, Q represents the maximum load of the trolley, *n* is the total number of production points, *g* represents the demand of each production point, the travelling distance of the trolley is represented by *D*, and the solution *D* of the shortest path is set as the objective function. Finally, the mathematical model of material scheduling can be obtained as follows:(24)$$minD=\overset{n}{\underset{i=0}{\sum }}\overset{n}{\underset{j=0}{\sum }}\overset{k}{\underset{v=0}{\sum }}c_{ij}x_{ijv}$$
(25)$$\overset{n}{\underset{i=0}{\sum }}x_{ijv}{=y}_{jv},j=1,2,\ldots ,n;v=1,2,\ldots ,k$$
(26)$$\overset{n}{\underset{j=0}{\sum }}x_{ijv}{=y}_{jv},i=1,2,\ldots ,n;v=1,2,\ldots ,k$$
(27)$$\overset{k}{\underset{v=0}{\sum }}y_{iv}=1,i=1,2,\ldots ,n$$
(28)$$\overset{n}{\underset{i=0}{\sum }}g_{i}y_{iv}\leq Q,v=1,2,\ldots ,k$$

In the mathematical model of the AGV material scheduling, minD denotes the shortest path of the objective function. Equations (25) and (26) ensure that the work can be accomplished at the production point. Equation (27) ensures that only one vehicle can complete each material point. Equation (28) ensures that the loaded weight of each cart cannot exceed the maximum loading capacity, which is the most important constraint for AGV material scheduling.

### 5.2. Experiment and Result Analysis

To begin, we tested with a small set of data. Table 14 shows the coordinates and requirements for the task points. In this short collection of test scenarios, the material point is 0, and the production point is 1–7. The maximum load is 100, and three vehicles finish the task allocation. The number of iterations in the test is 200, and the population size is 180, divided into six groups. The optimal path estimated by HP_PPE is 217.8, indicating that HP_PPE can optimize material scheduling.

Second, to further examine the role of HP_PPE in AGV workshop material scheduling, we selected seven sets of test data from VRPLIB, an international standard example for constrained vehicle routing scheduling (CVRP), and compared the test results with the PSO, EO, and PEO algorithms. In order to ensure the fairness of the results, the iteration times of the four algorithms were uniformly set to 3000 times, and the total particle number was 180. Table 15 records the best results of the four algorithms on seven sets of test data. Figure 5 shows selected and plotted change curves of the four algorithms on six test datasets.

By analyzing the data in Table 15 and comparing the convergence results of these four algorithms under the same experimental conditions, it can be found that HP_PPE can obtain a better convergence value than the PSO, EO, and PEO algorithms. As can be seen from Figure 5, compared with other algorithms, the convergence accuracy of HP_PPE is better than the other three algorithms. Although it cannot converge stably in the early stage, HP_PPE can jump out of the local optimum and achieve a better convergence value than other algorithms in the middle and late stages.

Finally, the algorithm was tested with a set of complex production shop data. The coordinates and requirements of the tasks are shown in Table 16, where zero is the coordinate of the material point and 1–10 are the coordinates of the production machines. The maximum load of the vehicle was set at 100, and the task allocation was completed by five vehicles. In the test, the number of iterations was 500, the population size was 180, and HP_PPE was divided into 6 groups. HP_PPE was compared with other algorithms, and the comparison results are shown in Table 17.

By analyzing the data in Table 17 under the same experimental conditions and comparing the best values, it can be found that HP_PPE can obtain a better solution than the PSO, EO, and PEO algorithms in material scheduling and is more likely to jump out of the local optimum. By comparing the average value, it can be found that the convergence accuracy of HP_PPE is higher than that of other algorithms. Due to the high complexity of HP_PPE itself, although a better result can be obtained, the running time of the algorithm is relatively long.

## 6. Conclusions

In this paper, the parallel and hybrid improvement strategies are applied. Firstly, the phasmatodea population evolution algorithm is mixed with the equilibrium optimization algorithm to enhance the ability of the algorithm to jump out of the local optimum and improve the convergence accuracy of the algorithm. On this basis, a new hybrid parallel balancing phasmatodea population evolution algorithm (HP_PPE) is proposed by using parallel mechanisms to shorten the algorithm’s running time. By comparing HP_PPE with six standard algorithms and then with three unique parallel algorithms by CEC2017, it can be found that the convergence accuracy of HP_PPE is better than most of the existing classical and new algorithms. It is faster than other algorithms in terms of convergence speed and better than other algorithms in terms of jumping out of the local optimum. In order to test the optimization ability of HP_PPE in a realistic scenario, the AGV workshop material scheduling model is selected in this paper, and the HP_PPE algorithm is compared with other algorithms in this scenario. The comparison results show that HP_PPE can obtain better results than other algorithms, which further proves the comprehensive optimization ability of the proposed HP_PPE algorithm and its ability to solve the material scheduling problem in the AGV workshop.

Since PPE was proposed recently, not much research has been conducted on it. Among the existing studies, there are hybrid approaches to hybridize it with other algorithms, or parallel approaches are used. In this paper, for the first time, both hybrid and parallel approaches are used to research PPE, and the results show that the proposed HP_PPE not only far outperforms other algorithms in the latest benchmark test suite CEC2017, but also significantly outperforms other algorithms in workshop material scheduling. Therefore, the research results of this paper are successful and impressive. Although the HP_PPE algorithm proposed in this paper has significantly improved the convergence speed and the ability to jump out of the local optimum compared with other algorithms, the improvement of the running time of this algorithm is not significant, which can be used as a direction for further improvement in the future. In addition, this algorithm can also be applied to problems in wireless sensor networks, public bus scheduling systems, feature selection, deep learning, and other fields.

## Figures and Tables

**Figure 1 entropy-25-00848-f001:**
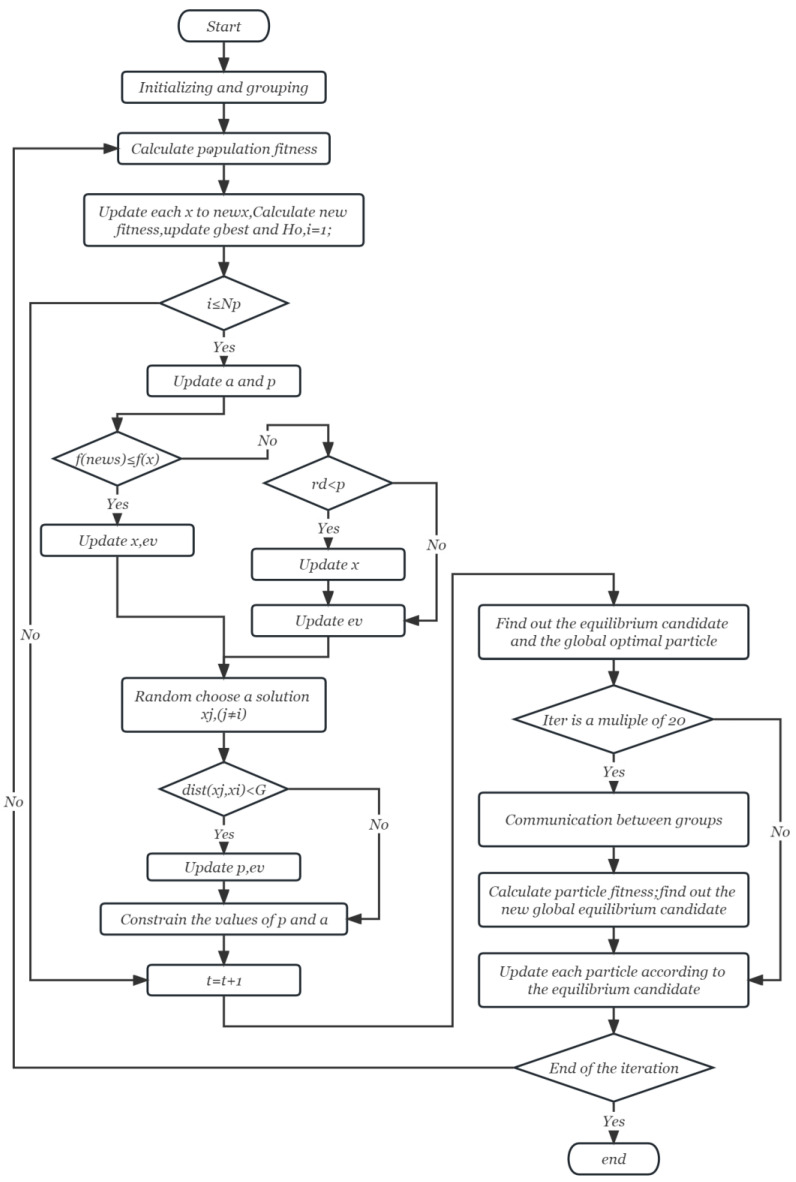
Flowchart of HP_PPE.

**Figure 2 entropy-25-00848-f002:**
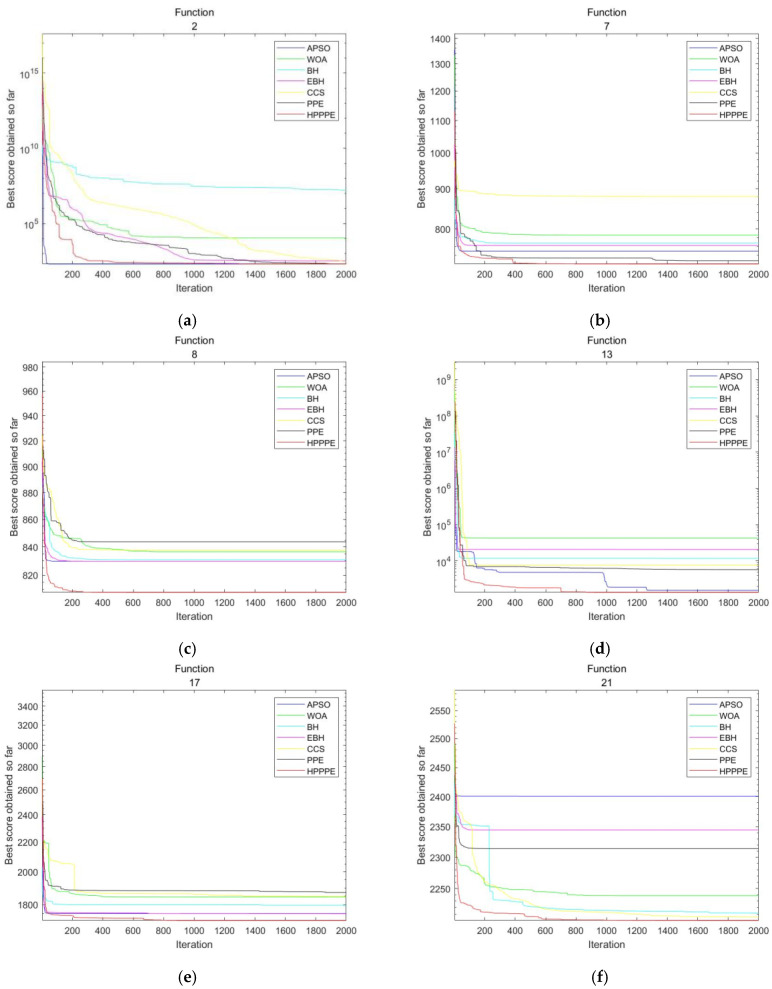
Convergence test results in 10 dimensions. (**a**–**n**): F2, F4, F6, F7, F8, F9, F13, F14, F17, F21, F22, F26, F28, F30.

**Figure 3 entropy-25-00848-f003:**
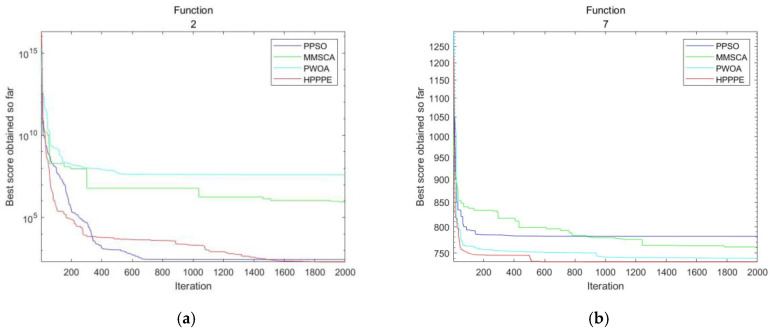
Convergence test results in 30 dimensions. (**a**–**r**): F2, F4, F7, F8, F9, F10, F11, F12, F14, F17, F18, F19, F22, F24, F25, F26, F28, F29.

**Figure 4 entropy-25-00848-f004:**
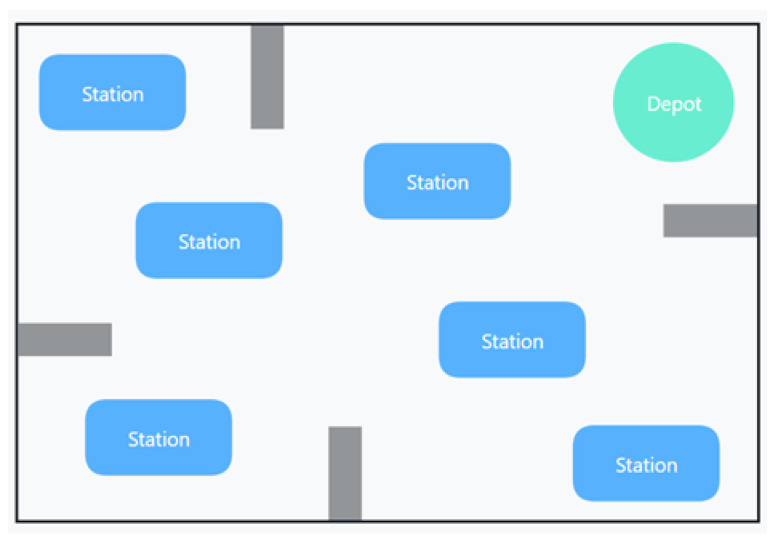
2-D workshop environment diagram.

**Figure 5 entropy-25-00848-f005:**
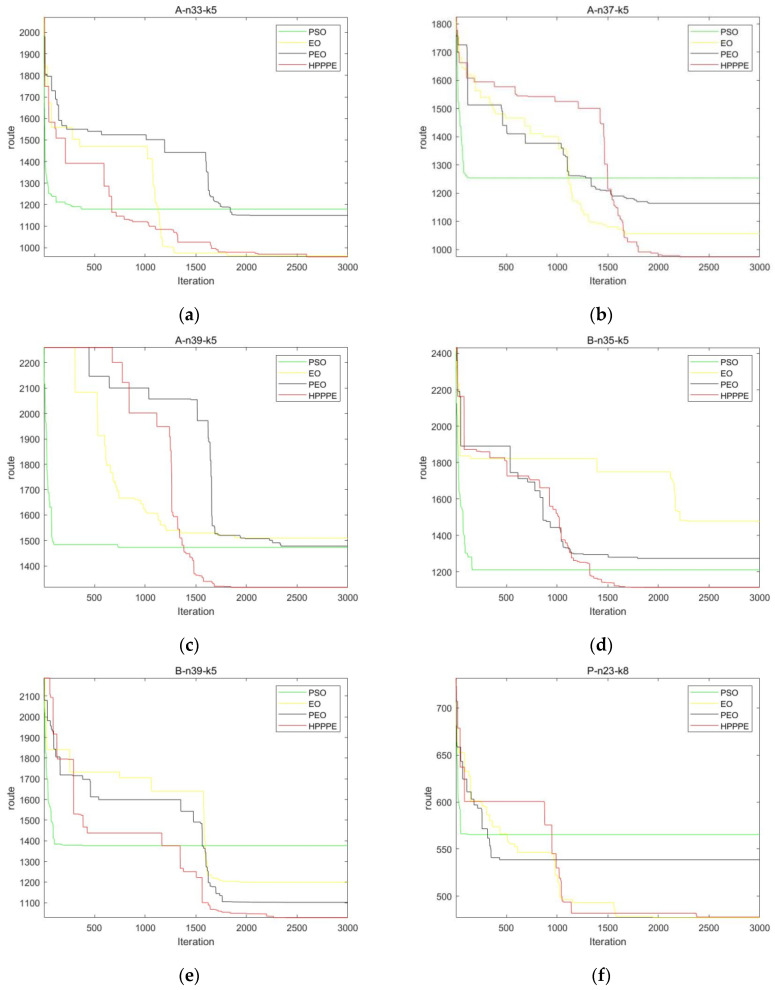
Convergence test results on six sets of test data. (**a**–**f**): A-n33-k5, A-n37-k5, A-n39-k5,B-n35-k5, B-n39-k5, P-n23-k8.

**Table 1 entropy-25-00848-t001:** Benchmark functions of CEC2017.

No.	Type	Functions
F1	Unimodal Functions	Shifted and Rotated Bent Cigar Function
F2		Shifted and Rotated Sum of Different Power Function *
F3		Shifted and Rotated Zakharov Function
F4	Simple Multimodal Functions	Shifted and Rotated Rosenbrock’s Function
F5		Shifted and Rotated Rastrigin’s Function
F6		Shifted and Rotated Expanded Scaffer’s F6 Function
F7		Shifted and Rotated Lunacek Bi_Rastrigin Function
F8		Shifted and Rotated Non-Continuous Rastrigin’s Function
F9		Shifted and Rotated Levy Function
F10		Shifted and Rotated Schwefel’s Function
F11	Hybrid Functions	Hybrid Function 1 (N = 3)
F12		Hybrid Function 2 (N = 3)
F13		Hybrid Function 3 (N = 3)
F14		Hybrid Function 4 (N = 4)
F15		Hybrid Function 5 (N = 4)
F16		Hybrid Function 6 (N = 4)
F17		Hybrid Function 6 (N = 5)
F18		Hybrid Function 6 (N = 5)
F19		Hybrid Function 6 (N = 5)
F20		Hybrid Function 6 (N = 6)
F21	Composition Functions	Composition Function 1 (N = 3)
F22		Composition Function 2 (N = 3)
F23		Composition Function 3 (N = 4)
F24		Composition Function 4 (N = 4)
F25		Composition Function 5 (N = 5)
F26		Composition Function 6 (N = 5)
F27		Composition Function 7 (N = 6)
F28		Composition Function 8 (N = 6)
F29		Composition Function 9 (N = 3)
F30		Composition Function 10 (N = 3)

* F2 exhibits unstable behavior.

**Table 2 entropy-25-00848-t002:** Comparison of average of fitness functions on 10D optimization among APSO, WOA, BH, EBH, CCS, PPE, and HP_PPE.

F(x)	APSO		WOA		BH		EBH		CCS		PPE		HP_PPE
F1	**2.5780 × 10³**	>	6.2338 × 10^6^	<	5.3197 × 10^8^	<	4.8922 × 10^3^	>	1.5447 × 10^4^	>	5.5384 × 10^3^	>	6.6144 × 10^4^
F2	**200.0024**	>	5.0351 × 10^5^	<	1.0925 × 10^7^	<	233.4492	<	2.4830 × 10^3^	<	210.3234	>	221.7403
F3	300.0551	>	2.7675 × 10^3^	<	2.9433 × 10^3^	<	**300.0000**	>	410.6009	<	300.0253	>	304.1250
F4	**401.9578**	>	434.0236	<	458.8530	<	405.0587	<	411.3768	<	407.0841	<	404.6802
F5	556.9483	<	559.3323	<	550.4430	<	536.8469	<	546.8702	<	**529.8673**	>	530.7990
F6	607.7350	<	635.5455	<	630.7529	<	623.7923	<	631.1905	<	602.5282	<	**601.6631**
F7	743.5262	<	784.6253	<	759.5797	<	760.6150	<	803.7964	<	727.5505	<	**726.1186**
F8	846.2986	<	843.1856	<	831.9381	<	831.7152	<	833.7346	<	819.1048	<	**816.0479**
F9	1.1640 × 10^3^	<	1.4532 × 10^3^	<	1.0743 × 10^3^	<	1.1991 × 10^3^	<	1.4386 × 10^3^	<	922.7870	<	**906.2824**
F10	2.1503 × 10^3^	<	2.1505 × 10^3^	<	2.2245 × 10^3^	<	2.0846 × 10^3^	<	2.0511 × 10^3^	<	1.8003 × 10^3^	<	**1.7245 × 10^3^**
F11	1.1388 × 10^3^	<	1.2071 × 10^3^	<	1.1764 × 10^3^	<	1.1967 × 10^3^	<	1.2134 × 10^3^	<	1.1251 × 10^3^	<	**1.1234 × 10^3^**
F12	1.7411 × 10^4^	<	3.8184 × 10^6^	<	1.2980 × 10^6^	<	9.4809 × 10^5^	<	2.2751 × 10^3^	<	2.5444 × 10^4^	<	**1.6041 × 10^4^**
F13	**4.8410 × 10^3^**	>	2.0652 × 10^4^	<	1.4149 × 10^4^	<	1.3826 × 10^4^	<	1.9960 × 10^4^	<	9.2302 × 10^3^	<	4.9364 × 10^3^
F14	**1.4488 × 10^3^**	>	1.9297 × 10^3^	<	2.9113 × 10^3^	<	1.7254 × 10^3^	>	1.6755 × 10^3^	>	1.4703 × 10^3^	>	1.9291 × 10^3^
F15	**1.5143 × 10^3^**	>	8.0922 × 10^3^	<	1.1019 × 10^4^	<	3.7836 × 10^3^	<	3.7061 × 10^3^	<	1.5514 × 10^3^	>	2.3897 × 10^3^
F16	1.9080 × 10^3^	<	1.8771 × 10^3^	<	1.8719 × 10^3^	<	1.8073 × 10^3^	<	**1.7789 × 10^3^**	>	1.8437 × 10^3^	<	1.7908 × 10^3^
F17	1.7950 × 10^3^	<	1.8162 × 10^3^	<	1.7939 × 10^3^	<	1.7984 × 10^3^	<	1.7771 × 10^3^	<	1.7567 × 10^3^	<	**1.7509 × 10^3^**
F18	9.1038 × 10^3^	<	1.5625 × 10^4^	<	8.2555 × 10^3^	<	2.5033 × 10^4^	<	3.2619 × 10^4^	<	1.1399 × 10^4^	<	**5.4399 × 10^3^**
F19	2.9469 × 10^3^	>	4.4361 × 10^4^	<	7.9903 × 10^3^	<	4.9336 × 10^3^	<	4.2507 × 10^3^	<	**2.1573 × 10^3^**	>	3.7627 × 10^3^
F20	2.1158 × 10^3^	<	2.1927 × 10^3^	<	2.1017 × 10^3^	<	2.1515 × 10^3^	<	2.1097 × 10^3^	<	2.0706 × 10^3^	<	**2.0569 × 10^3^**
F21	2.3436 × 10^3^	<	2.3232 × 10^3^	<	2.2295 × 10^3^	>	2.2481 × 10^3^	>	**2.2034 × 10^3^**	>	2.3021 × 10^3^	<	2.2902 × 10^3^
F22	2.6464 × 10^3^	<	2.4197 × 10^3^	<	2.3414 × 10^3^	<	2.3082 × 10^3^	<	2.3136 × 10^3^	<	**2.3043 × 10^3^**	>	2.3049 × 10^3^
F23	2.7347 × 10^3^	<	2.6526 × 10^3^	<	2.6798 × 10^3^	<	2.6444 × 10^3^	<	2.6579 × 10^3^	<	2.6499 × 10^3^	<	**2.6418 × 10^3^**
F24	2.8222 × 10^3^	<	2.7751 × 10^3^	<	2.6583 × 10^3^	>	2.7209 × 10^3^	<	**2.5215 × 10^3^**	>	2.7280 × 10^3^	<	2.7207 × 10^3^
F25	2.9215 × 10^3^	<	2.9345 × 10^3^	<	2.9479 × 10^3^	<	2.9357 × 10^3^	<	2.9351 × 10^3^	<	2.9295 × 10^3^	<	**2.9096 × 10^3^**
F26	3.3712 × 10^3^	<	3.6168 × 10^3^	<	3.1596 × 10^3^	<	3.0091 × 10^3^	<	3.1325 × 10^3^	<	2.9663 × 10^3^	<	**2.8676 × 10^3^**
F27	3.1906 × 10^3^	<	3.1378 × 10^3^	<	3.1685 × 10^3^	<	**3.1106 × 10^3^**	>	3.1129 × 10^3^	>	3.1382 × 10^3^	<	3.1372 × 10^3^
F28	3.3782 × 10^3^	<	3.4140 × 10^3^	<	3.2421 × 10^3^	<	3.2967 × 10^3^	<	**3.2058 × 10^3^**	>	3.3009 × 10^3^	<	3.2269 × 10^3^
F29	3.3472 × 10^3^	<	3.3964 × 10^3^	<	3.2738 × 10^3^	<	3.2740 × 10^3^	<	3.2380 × 10^3^	<	3.2542 × 10^3^	<	**3.2332 × 10^3^**
F30	3.7910 × 10^5^	<	1.3732 × 10^6^	<	1.3877 × 10^6^	<	9.6717 × 10^5^	<	2.8117 × 10^5^	<	2.6690 × 10^5^	<	**1.9652 × 10^5^**
</=/>	22/0/8		30/0/0		28/0/2		25/0/5		23/0/7		22/0/8		-

**Table 3 entropy-25-00848-t003:** Comparison of average of fitness functions on 30D optimization among APSO, WOA, BH, EBH, CCS, PPE, and HP_PPE.

F(x)	APSO		WOA		BH		EBH		CCS		PPE		HP_PPE
F1	8.7770 × 10^3^	>	1.1805 × 10^9^	<	1.2814 × 10^10^	<	**4.0908 × 10^3^**	>	5.0862 × 10^6^	<	2.0950 × 10^5^	>	4.1310 × 10^6^
F2	**218.8960**	>	5.0621 × 10^32^	<	3.8516 × 10^40^	<	1.1854 × 10^15^	<	3.7737 × 10^31^	<	2.2261 × 10^12^	>	1.3860 × 10^13^
F3	**4.5606 × 10^3^**	>	2.3633 × 10^5^	<	7.3016 × 10^4^	<	7.1105 × 10^3^	>	8.7552 × 10^4^	<	7.2491 × 10^3^	>	1.1754 × 10^4^
F4	**482.3065**	>	774.8969	<	3.0501 × 10^3^	<	494.6746	>	530.0303	<	508.6853	>	522.9445
F5	792.5104	<	833.0313	<	773.8702	<	723.7200	<	785.4899	<	677.8883	<	**675.6955**
F6	**623.5470**	>	679.6597	<	671.2125	<	656.3925	<	668.3710	<	642.0989	<	640.1540
F7	**964.6982**	>	1.2737 × 10^3^	<	1.1831 × 10^3^	<	1.1591 × 10^3^	<	1.3495 × 10^3^	<	976.2660	>	977.9216
F8	1.0424 × 10^3^	<	1.0391 × 10^3^	<	1.0299 × 10^3^	<	976.9357	<	987.3624	<	939.2946	<	**937.0133**
F9	7.6037 × 10^3^	<	1.0526 × 10^4^	>	6.7611 × 10^3^	<	5.3157 × 10^3^	<	6.8942 × 10^3^	<	**4.2115 × 10^3^**	>	4.6635 × 10^3^
F10	5.2695 × 10^3^	<	7.0705 × 10^3^	<	7.6673 × 10^3^	<	5.5310 × 10^3^	<	5.6721 × 10^3^	<	4.9834 × 10^3^	<	**4.7256 × 10^3^**
F11	**1.2273 × 10^3^**	>	5.2209 × 10^3^	<	2.6162 × 10^3^	<	1.3134 × 10^3^	<	1.5067 × 10^3^	<	1.2336 × 10^3^	>	1.2475 × 10^3^
F12	**4.2016 × 10^5^**	>	1.9182 × 10^8^	<	2.2066 × 10^9^	<	1.2938 × 10^7^	<	3.7364 × 10^7^	<	2.2810 × 10^6^	<	1.7365 × 10^6^
F13	**1.5019 × 10^4^**	>	1.1449 × 10^6^	<	5.3896 × 10^8^	<	1.2394 × 10^5^	<	1.6483 × 10^5^	<	3.3412 × 10^4^	<	3.3070 × 10^4^
F14	3.0582 × 10^4^	>	2.5465 × 10^6^	<	4.6671 × 10^5^	<	4.3822 × 10^4^	>	2.8609 × 10^5^	<	**1.4801 × 10^4^**	>	7.8681 × 10^4^
F15	1.1475 × 10^4^	<	1.1390 × 10^6^	<	2.0054 × 10^4^	<	4.8117 × 10^4^	<	5.4937 × 10^4^	<	7.6309 × 10^3^	<	**3.3261 × 10^3^**
F16	3.0370 × 10^3^	<	4.2136 × 10^3^	<	4.2478 × 10^3^	<	3.4134 × 10^3^	<	3.6435 × 10^3^	<	2.8134 × 10^3^	<	**2.6949 × 10^3^**
F17	2.4174 × 10^3^	<	2.6527 × 10^3^	<	2.7703 × 10^3^	<	2.4361 × 10^3^	<	2.8097 × 10^3^	<	2.2966 × 10^3^	<	**2.2524 × 10^3^**
F18	**2.4008 × 10^5^**	>	8.9304 × 10^6^	<	8.8726 × 10^5^	<	6.8510 × 10^5^	<	3.2000 × 10^6^	<	2.5478 × 10^5^	>	3.6297 × 10^5^
F19	1.0555 × 10^4^	<	9.1050 × 10^6^	<	9.5380 × 10^5^	<	5.2232 × 10^5^	<	3.8705 × 10^6^	<	5.6581 × 10^3^	<	**5.5680 × 10^3^**
F20	2.7245 × 10^3^	<	2.8685 × 10^3^	<	2.6924 × 10^3^	<	2.7382 × 10^3^	<	2.6837 × 10^3^	<	2.6208 × 10^3^	<	**2.6040 × 10^3^**
F21	2.6034 × 10^3^	<	2.6145 × 10^3^	<	2.6047 × 10^3^	<	2.5087 × 10^3^	<	2.5752 × 10^3^	<	**2.4621 × 10^3^**	>	2.4753 × 10^3^
F22	7.1563 × 10^3^	<	7.8469 × 10^3^	<	6.5478 × 10^3^	<	5.6361 × 10^3^	<	6.9669 × 10^3^	<	**4.4150 × 10^3^**	>	4.5172 × 10^3^
F23	3.4232 × 10^3^	<	3.0807 × 10^3^	>	3.3474 × 10^3^	<	**3.0424 × 10^3^**	>	3.1508 × 10^3^	<	3.0993 × 10^3^	>	3.1144 × 10^3^
F24	3.5234 × 10^3^	<	3.1915 × 10^3^	>	3.5856 × 10^3^	<	**3.1820 × 10^3^**	>	3.3307 × 10^3^	<	3.2577 × 10^3^	<	3.2302 × 10^3^
F25	**2.8990 × 10^3^**	>	3.0769 × 10^3^	<	3.1925 × 10^3^	<	2.9199 × 10^3^	>	2.9463 × 10^3^	<	2.9285 × 10^3^	>	2.9323 × 10^3^
F26	7.6952 × 10^3^	<	8.2106 × 10^3^	<	8.3646 × 10^3^	<	6.7678 × 10^3^	<	7.7051 × 10^3^	<	5.6433 × 10^3^	<	**5.1623 × 10^3^**
F27	3.4687 × 10^3^	>	3.4352 × 10^3^	>	4.1485 × 10^3^	<	3.3944 × 10^3^	>	**3.3005 × 10^3^**	>	3.6061 × 10^3^	<	3.4742 × 10^3^
F28	**3.1710 × 10^3^**	>	3.4867 × 10^3^	<	4.1489 × 10^3^	<	3.2589 × 10^3^	>	3.3250 × 10^3^	<	3.2730 × 10^3^	<	3.2640 × 10^3^
F29	4.3101 × 10^3^	<	5.2795 × 10^3^	<	5.6814 × 10^3^	<	4.6004 × 10^3^	<	5.0842 × 10^3^	<	4.1059 × 10^3^	<	**4.0950 × 10^3^**
F30	**1.0730 × 10^4^**	>	3.1793 × 10^7^	<	1.7950 × 10^7^	<	3.5444 × 10^6^	<	9.5391 × 10^6^	<	1.0825 × 10^5^	<	2.2837 × 10^4^
</=/>	15/0/15		26/0/4		30/0/0		21/0/9		29/0/1		17/0/13		-

**Table 4 entropy-25-00848-t004:** Performance of different algorithms on different types of functions (10D).

Algorithm	Unimodal	Multimodal	Hybrid	Composition	Win
HP_PPE	0	5	5	5	15
APSO	2	1	3	0	6
CCS	0	0	1	3	4
PPE	0	1	1	1	3
EBH	1	0	0	1	2
WOA	0	0	0	0	0
BH	0	0	0	0	0

**Table 5 entropy-25-00848-t005:** Performance of different algorithms on different types of functions (30D).

Algorithm	Unimodal	Multimodal	Hybrid	Composition	Win
APSO	2	3	4	3	12
HP_PPE	0	3	5	2	10
PPE	0	1	1	2	4
EBH	1	0	0	2	3
CCS	0	0	0	1	1
WOA	0	0	0	0	0
BH	0	0	0	0	0

**Table 6 entropy-25-00848-t006:** Wilcoxon signed rank test results (10D).

Comparison	R+	R−	*p*-Value	Sig.
HP_PPE versus APSO	302	163	0.1529	≈
HP_PPE versus WOA	462	3	2.3534 × 10^−6^	+
HP_PPE versus BH	447	18	1.0246 × 10^−5^	+
HP_PPE versus EBH	374	91	0.0036	+
HP_PPE versus CCS	360	105	0.0087	+
HP_PPE versus PPE	282	183	0.3086	≈

**Table 7 entropy-25-00848-t007:** Wilcoxon signed rank test results (30D).

Comparison	R+	R−	*p*-Value	Sig.
HP_PPE versus APSO	231	234	0.9754	≈
HP_PPE versus WOA	459	6	3.1817 × 10^−6^	+
HP_PPE versus BH	465	0	1.7344 × 10^−6^	+
HP_PPE versus EBH	337	128	0.0316	+
HP_PPE versus CCS	454	11	5.2165 × 10^−6^	+
HP_PPE versus PPE	212	253	0.4217	≈

**Table 8 entropy-25-00848-t008:** Comparison of average of fitness functions on 10D optimization among PPSO, MMSCA, and PWOA.

F(x)	PPSO		MMSCA		PWOA		HP_PPE
F1	**1.8595 × 10^3^**	>	2.4656 × 10^8^	<	4.2927 × 10^7^	<	6.6144 × 10^4^
F2	224.2367	<	3.8383 × 10^5^	<	7.3928 × 10^6^	<	**221.7403**
F3	**300.1454**	>	606.7382	<	2.0033 × 10^3^	<	304.1250
F4	408.7630	<	418.4225	<	420.9497	<	**404.6802**
F5	559.6310	<	533.1216	<	550.2127	<	**530.7990**
F6	635.8977	<	610.7840	<	623.9905	<	**601.6631**
F7	751.0268	<	754.6577	<	770.2714	<	**726.1186**
F8	831.2424	<	825.7253	<	833.0629	<	**816.0479**
F9	1.1617 × 10^3^	<	932.0829	<	1.3119 × 10^3^	<	**906.2824**
F10	2.2004 × 10^3^	<	1.7965 × 10^3^	<	2.0385 × 10^3^	<	**1.7245 × 10^3^**
F11	1.1491 × 10^3^	<	1.1455 × 10^3^	<	1.1917 × 10^3^	<	**1.1234 × 10^3^**
F12	**1.2447 × 10^4^**	>	2.0371 × 10^6^	<	3.0906 × 10^6^	<	1.6041 × 10^4^
F13	**2.8311 × 10^3^**	>	6.1275 × 10^3^	<	1.1159 × 10^4^	<	4.9364 × 10^3^
F14	**1.4867 × 10^3^**	>	1.4878 × 10^3^	>	1.9257 × 10^3^	>	1.9291 × 10^3^
F15	1.6374 × 10^3^	>	**1.6298 × 10^3^**	>	4.8050 × 10^3^	<	2.3897 × 10^3^
F16	1.8739 × 10^3^	<	**1.6490 × 10^3^**	>	1.8140 × 10^3^	<	1.7908 × 10^3^
F17	1.7782 × 10^3^	<	1.7540 × 10^3^	<	1.7800 × 10^3^	<	**1.7509 × 10^3^**
F18	6.5632 × 10^3^	<	3.0830 × 10^4^	<	2.0578 × 10^4^	<	**5.4399 × 10^3^**
F19	2.8965 × 10^3^	>	**2.0029 × 10^3^**	>	1.2560 × 10^4^	<	3.7627 × 10^3^
F20	2.1749 × 10^3^	<	2.0614 × 10^3^	<	2.1420 × 10^3^	<	**2.0569 × 10^3^**
F21	2.3201 × 10^3^	<	**2.2051 × 10^3^**	>	2.3179 × 10^3^	<	2.2902 × 10^3^
F22	2.3500 × 10^3^	<	2.3140 × 10^3^	<	2.3601 × 10^3^	<	**2.3049 × 10^3^**
F23	2.7285 × 10^3^	<	**2.6409 × 10^3^**	>	2.6530 × 10^3^	<	2.6418 × 10^3^
F24	2.6966 × 10^3^	>	**2.6416 × 10^3^**	>	2.7428 × 10^3^	<	2.7207 × 10^3^
F25	2.9203 × 10^3^	<	2.9234 × 10^3^	<	2.9517 × 10^3^	<	**2.9096 × 10^3^**
F26	3.1147 × 10^3^	<	3.0036 × 10^3^	<	3.1109 × 10^3^	<	**2.8676 × 10^3^**
F27	3.1758 × 10^3^	<	**3.0988 × 10^3^**	>	3.1244 × 10^3^	>	3.1372 × 10^3^
F28	3.3247 × 10^3^	<	**3.2075 × 10^3^**	>	3.4084 × 10^3^	<	3.2269 × 10^3^
F29	3.2877 × 10^3^	<	**3.1801 × 10^3^**	>	3.3257 × 10^3^	<	3.2332 × 10^3^
F30	1.3451 × 10^5^	>	**9.2503 × 10^4^**	>	2.0727 × 10^5^	<	1.9652 × 10^5^
</=/>	21/0/9		19/0/11		28/0/2		-

**Table 9 entropy-25-00848-t009:** Comparison of average of fitness functions on 30D optimization among PPSO, MMSCA, and PWOA.

F(x)	PPSO		MMSCA		PWOA		HP_PPE
F1	**3.4046 × 10^6^**	>	1.0650 × 10^10^	<	4.8333 × 10^9^	<	4.1310 × 10^6^
F2	4.5920 × 10^17^	<	2.0586 × 10^32^	<	2.1339 × 10^32^	<	**1.3860 × 10^13^**
F3	**6.7412 × 10^3^**	>	3.3366 × 10^4^	<	1.6978 × 10^5^	<	1.1754 × 10^4^
F4	**513.5943**	>	1.1586 × 10^3^	<	962.2483	<	522.9445
F5	732.4175	<	758.3829	<	796.3011	<	**675.6955**
F6	661.4341	<	643.4638	<	666.1981	<	**640.1540**
F7	1.0793 × 10^3^	<	1.0897 × 10^3^	<	1.2526 × 10^3^	<	**977.9216**
F8	982.1570	<	1.0315 × 10^3^	<	1.0182 × 10^3^	<	**937.0133**
F9	5.3564 × 10^3^	<	**4.3691 × 10^3^**	>	7.0559 × 10^3^	<	4.6635 × 10^3^
F10	5.3147 × 10^3^	<	7.7807 × 10^3^	<	6.5770 × 10^3^	<	**4.7256 × 10^3^**
F11	1.2880 × 10^3^	<	1.8636 × 10^3^	<	3.9675 × 10^3^	<	**1.2475 × 10^3^**
F12	1.4607 × 10^3^	<	8.4850 × 10^8^	<	2.7355 × 10^8^	<	**1.7365 × 10^3^**
F13	9.8003 × 10^4^	<	2.3672 × 10^8^	<	8.0933 × 10^6^	<	**3.3070 × 10^4^**
F14	**1.0725 × 10^4^**	>	7.6207 × 10^4^	>	1.2822 × 10^6^	<	7.8681 × 10^4^
F15	2.9877 × 10^4^	<	4.0385 × 10^6^	<	5.6747 × 10^6^	<	**3.3261 × 10^3^**
F16	3.0331 × 10^3^	<	3.4025 × 10^3^	<	3.4872 × 10^3^	<	**2.6949 × 10^3^**
F17	2.4942 × 10^3^	<	2.2533 × 10^3^	<	2.6007 × 10^3^	<	**2.2524 × 10^3^**
F18	**1.5905 × 10^5^**	>	1.9971 × 10^6^	<	3.5745 × 10^6^	<	3.6297 × 10^5^
F19	1.1781 × 10^5^	<	1.1985 × 10^7^	<	2.2354 × 10^6^	<	**5.5680 × 10^3^**
F20	2.8214 × 10^3^	<	**2.4882 × 10^3^**	>	2.8115 × 10^3^	<	2.6040 × 10^3^
F21	2.5449 × 10^3^	<	2.5371 × 10^3^	<	2.5838 × 10^3^	<	**2.4753 × 10^3^**
F22	6.0040 × 10^3^	<	**3.6261 × 10^3^**	>	6.5017 × 10^3^	<	4.5172 × 10^3^
F23	3.3470 × 10^3^	<	**2.9525 × 10^3^**	>	3.0576 × 10^3^	>	3.1144 × 10^3^
F24	3.3122 × 10^3^	<	**3.1304 × 10^3^**	>	3.1685 × 10^3^	>	3.2302 × 10^3^
F25	2.9476 × 10^3^	<	3.1492 × 10^3^	<	3.1142 × 10^3^	<	**2.9323 × 10^3^**
F26	6.1899 × 10^3^	<	6.5017 × 10^3^	<	7.8819 × 10^3^	<	**5.1623 × 10^3^**
F27	3.5951 × 10^3^	<	**3.3662 × 10^3^**	>	3.4154 × 10^3^	>	3.4742 × 10^3^
F28	3.2729 × 10^3^	<	3.6907 × 10^3^	<	3.6906 × 10^3^	<	**3.2640 × 10^3^**
F29	4.7297 × 10^3^	<	4.3966 × 10^3^	<	4.8452 × 10^3^	<	**4.0950 × 10^3^**
F30	1.6738 × 10^6^	<	4.0668 × 10^7^	<	1.2036 × 10^7^	<	**2.2837 × 10^4^**
</=/>	25/0/5		23/0/7		27/0/3		-

**Table 10 entropy-25-00848-t010:** Performance of different algorithms on different types of parallel functions (10D).

Algorithm	Unimodal	Multimodal	Hybrid	Composition	Win
HP_PPE	1	7	4	3	15
MMSCA	0	0	3	7	10
PPSO	2	0	3	0	5
PWOA	0	0	0	0	0

**Table 11 entropy-25-00848-t011:** Performance of different algorithms on different types of parallel functions (30D).

Algorithm	Unimodal	Multimodal	Hybrid	Composition	Win
HP_PPE	1	5	7	6	19
MMSCA	0	1	1	4	6
PPSO	2	1	2	0	5
PWOA	0	0	0	0	0

**Table 12 entropy-25-00848-t012:** Wilcoxon signed rank test results for different parallel algorithms (10D).

Comparison	R+	R−	*p*-Value	Sig.
HP_PPE versus PPSO	290	175	0.2369	≈
HP_PPE versus MMSCA	251	214	0.7036	≈
HP_PPE versus PWOA	455	10	4.7292 × 10^−6^	+

**Table 13 entropy-25-00848-t013:** Wilcoxon signed rank test results for different parallel algorithms (30D).

Comparison	R+	R-	*p*-Value	Sig.
HP_PPE versus PPSO	366	99	0.006	+
HP_PPE versus MMSCA	419	46	1.2506 × 10^4^	+
HP_PPE versus PWOA	450	15	7.6909 × 10^6^	+

**Table 14 entropy-25-00848-t014:** The task point coordinates and demand.

Sequence	0	1	2	3	4	5	6	7
Coordinate	(18,54)	(22,60)	(58,69)	(71,71)	(83,46)	(91,38)	(24,42)	(18,40)
Requirement	0	89	14	28	33	21	41	57

**Table 15 entropy-25-00848-t015:** The best results of the four algorithms on seven sets of test data.

Data	PSO-Route	EO-Route	PEO-Route	HP_PPE-Route
A-n33-k5	1.2037 × 10^3^	1.1153 × 10^3^	1.1434 × 10^3^	1.1072 × 10^3^
A-n37-k5	1.1118 × 10^3^	1.0788 × 10^3^	1.2195 × 10^3^	1.0758 × 10^3^
A-n39-k5	1.5895 × 10^3^	1.4251 × 10^3^	1.6289 × 10^3^	1.3870 × 10^3^
B-n35-k5	1.4140 × 10^3^	1.1735 × 10^3^	1.1848 × 10^3^	1.0751 × 10^3^
B-n39-k5	1.2859 × 10^3^	1.0835 × 10^3^	1.0481 × 10^3^	1.0460 × 10^3^
P-n20-k2	450.3385	493.5011	475.8411	449.5048
P-n23-k8	573.645	486.8819	508.2372	465.0878

**Table 16 entropy-25-00848-t016:** Coordinates and requirements of machines in the workshop.

Sequence	Coordinate	Requirement
0	(116.426549,39.779675)	0
1	(116.323645,39.961334)	4
2	(116.409614,39.942402)	6
3	(116.363324,39.976932)	3
4	(116.316225,39.936386)	11
5	(116.431244,39.986622)	10
6	(116.354304,40.006782)	5
7	(116.3259,39.930093)	3
8	(116.324696,39.845583)	4
9	(39.845583,39.986873)	6
10	(116.472828,39.988674)	2

**Table 17 entropy-25-00848-t017:** Comparison results of different algorithms.

	PSO-Route	EO-Route	PEO-Route	HP_PPE-Route
best	199.9061	100.1945	100.2055	100.1712
mean	199.9495	100.4263	100.4707	100.3385

## Data Availability

The CVRP public data supporting the conclusions of this paper are available at http://vrp.atd-lab.inf.puc-rio.br/ (accessed on 29 November 2022).

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
