# Peer review of "A Hybrid Parallel Balanced Phasmatodea Population Evolution Algorithm and Its Application in Workshop Material Scheduling"

_entropy, 2023, doi:10.3390/e25060848_

Round 1

Reviewer 1 Report

Overall comments:

The paper is well-formed and quite interesting to read. With some improvements, the paper can be a worthy addition to the technical literature.

Here are some suggestions for the improvement of the manuscript:

Major points:

[1] The authors mention that "The number of groups for parallel grouping in this article's HP_PPE is set to two." without explicitly providing any reason(s)/explanation for the same. It would be instructive to understand this choice of 'groups' and what effect different choices may have on the performance of the HP_PPE.

[2] Figure 1 (Flowchart of HP_PPE) needs two things: (i) a better and more elaborate caption, and (ii) a high-resolution image.

[3] Similarly, the caption of Figure 2. Which is "3-D map for 2-D function" needs revision to highlight the salient features of the figure. The current caption is not clear.

[4] It is not clear why the EO algorithm was chosen for the hybridisation of the PPE algorithm -- as there are various possible candidates for this choice. This needs to be clearly elucidated in the paper.

Minor Points:

[1] It is advised to not keep the headings of the sections/sub-sections as abbreviations (such as EO, PPE, etc.). It would be better to keep the expanded versions of these abbreviations in the headings.

[2] There is a comma (,) appearing after every equation. All these are redundant and should be removed.

[3] The 'a' in line 240 (growth rate a) should be italicised. 

Author Response

Response to Reviewer 1 Comments

Comments: The paper is well-formed and quite interesting to read. With some improvements, the paper can be a worthy addition to the technical literature.

Response: We greatly appreciate your constructive comments and we have prepared a detailed reply to your comments.

Major points:

point 1: The authors mention that "The number of groups for parallel grouping in this article's HP_PPE is set to two." without explicitly providing any reason(s)/explanation for the same. It would be instructive to understand this choice of 'groups' and what effect different choices may have on the performance of the HP_PPE.

Response 1: Thanks a lot for your comments. We added a description to set HP_PPE to two groups. It is rectified at Lines 323 – 327.

point 2: Figure 1 (Flowchart of HP_PPE) needs two things: (i) a better and more elaborate caption, and (ii) a high-resolution image.

Response 2: Thanks a lot for your comments. We replaced the original flowchart with a better and clearer flowchart. It is rectified at Line 359.

point 3: Similarly, the caption of Figure 2. Which is "3-D map for 2-D function" needs revision to highlight the salient features of the figure. The current caption is not clear.

Response 3: Thanks a lot for your comments. Since this diagram is time-consuming to draw and their existence does not matter much to Section 4.1, we choose to remove the diagram and its description. It was removed at Line 367 and Line 377.

point 4: It is not clear why the EO algorithm was chosen for the hybridization of the PPE algorithm -- as there are various possible candidates for this choice. This needs to be clearly elucidated in the paper.

Response 4: Thanks a lot for your comments. We have modified and explained the reasons for choosing EO as the hybrid algorithm for PPE. It was modified on Lines 233 – 244.

Minor points:

point 1: It is advised to not keep the headings of the sections/sub-sections as abbreviations (such as EO, PPE, etc.). It would be better to keep the expanded versions of these abbreviations in the headings.

Response 1: Thanks a lot for your comments. We have kept extended versions of these abbreviations in the title. It is rectified at Line 137 and Line 172.

point 2: There is a comma (,) appearing after every equation. All these are redundant and should be removed.

Response 2: Thanks a lot for your comments. The commas after all equations have been removed.

point 3: The 'a' in line 240 (growth rate a) should be italicized.

Response 3: Thanks a lot for your comments. It is rectified at Line 277.

  Once again, thank you very much for your suggestions. We would be glad to respond to any further questions and comments that you may have.

Reviewer 2 Report

In this paper, the balanced optimization algorithm is combined with the phasmatodea population evolution algorithm to improve its convergence speed and global search capability. The hybrid model is tested by the AGV workshop material scheduling problem. There are the main problems in this paper as follows.

1.      The hybrid model is the combination of two mature algorithms. Therefore, it lacks novelty.

2.      The introduction and application analysis of the hybrid algorithm are relatively detailed, but the innovation of this paper is insufficient.

3.      In the "Related work" section, the basic principle of PPE, EO, and AGV workshop material scheduling should not be introduced in detail.

Author Response

Response to Reviewer 2 Comments

Comments: In this paper, the balanced optimization algorithm is combined with the phasmatodea population evolution algorithm to improve its convergence speed and global search capability. The hybrid model is tested by the AGV workshop material scheduling problem.

Response: Thank you for your suggestions.All your suggestions are very important, and they are of great guiding significance to our scientific research.We would be glad to respond to any further questions and comments that you may have.

Point 1: The hybrid model is the combination of two mature algorithms. Therefore, it lacks novelty.

Response 1: Thanks a lot for your comments. The strategy used in this paper is a combination of PPE and EO and the group parallelism mechanism. The original paper of PPE was published in 2021 and the original paper of EO was published in 2019, and EO is indeed a more mature algorithm compared to PPE. It is worth mentioning that the comparison tests in the test suite show that the HP_PPE proposed in this paper exhibits an impressive performance. For this issue, we will try to incorporate more novel algorithms in our subsequent research.

Point 2: The introduction and application analysis of the hybrid algorithm are relatively detailed, but the innovation of this paper is insufficient.

Response 2: Thanks a lot for your comments. Our research combines both hybrid and grouped parallel strategies to construct HP_PPE and applies them to the workshop material scheduling problem. To our knowledge, there are articles on PPE that use hybrid or grouped-parallel approaches, however, our paper is the first to combine hybrid and grouped-parallel strategies for the study of PPE. We have further explained this. It is rectified at Lines 117 – 120.

Point 3: In the "Related work" section, the basic principle of PPE, EO, and AGV workshop material scheduling should not be introduced in detail.

Response 3: Thanks a lot for your comments. We have streamlined the introduction to the fundamentals in the ''Related Work'' section. At the request of the 4th reviewer, we have added the process of finding a solution to the introduction of the PPE. It is rectified at Lines 132 – 224.

  Once again, thank you very much for your suggestions. We would be glad to respond to any further questions and comments that you may have.

Reviewer 3 Report

The article entitled: "A Hybrid Parallel Balanced Phasmatodea Population Evolution Algorithm and Its Application in AGV Workshop Material Scheduling" concern hybrid parallel balanced phasmatodea population evolution algorithm to workshop material scheduling problem. The manuscript can be interesting due to application really new optimization algorithm Phasmatodea Population Evolution Algorithm and modification of this algorithm. At the beginning my review I must underline, that the reference list must be modified before final publication of the manuscript. The main reason of deep modification is big number of citation of co-author of this manuscript - Jeng-Shyang Pan. I suggest the authors citation some authors from out of China or Asia. At present many citations come from Asia. I suggest resignation from citation [4, 5] where in list of authors is Pan, J. S and cited manuscript Dervis Karaboga from Turkey. In the first sentence you can cited "Performance analysis of selected metaheuristic optimization algorithms applied in the solution of an unconstrained task" published in Compel in 2022. In case of ACO algorithm is also cited the manuscript Pan, J. S. (2004) - I suggest citation of Mehmet Cunkas "Color image segmentation based on multiobjective artificial bee colony optimization" published in Applied Soft Computing. In cas of Cuckoo Search algorithm - citation Pan, J. S - I suggest changes on manuscript written by Venkateswararao Bathina "Hybrid Approach with Combining Cuckoo-Search and Grey-Wolf Optimizer for Solving Optimal Power Flow Problems". Also some manuscript of Fausto Marquez can be taken into account, such as: "Wind integrated power system to reduce emission: An application of Bat algorithm" and "Allocation of real power generation based on computing over all generation cost: an approach of Salp Swarm Algorithm”.

The comments and remarks to the authors are listed below:

1.      The introduction should be modified. At this version the more detailed information about PPE algorithm. The authors should focus on the research presented in the manuscript.

2.      The authors should resigned from acronym AGV in the title of the manuscript.

3.      The introduction is well written. The authors should modified the cited references to avoid the self-citation of the one of co-authors.

4.      The explanation of the acronym AGV is not introduced in the manuscript body. According to me it can be "Automated Guided Vehicle".

5.      In the description of "equilibrium optimization algorithm", subsection use symbol iter and Max_iter. Maybe, the authots use shorter descriptions. Additionaly, in the Equation (7) the description Max_iter are different.

6.      The same symbols should be mean the same parameters. For example in Equation (10), " ?1 and ?2 are two random numbers", but in Equation (4) the authors use rand, and in Equation (3) r1 denote " ?1 is the population's effective growth rate". The description of the symbols are untidy.

7.      In title of section "2.3.AGV workshop material scheduling" is lack of space before AGV.

8.      The authors use different font in the text and in equations. The symbols should be unified.

9.      In the row 287, there are "Ceq1,Ceq2,Ceq3and Ceq4" and should be written by italic font.

10.  In the Equation (23) the authors use "x" to multiplication. In the mathematics theory, the symbol "x" is used to vector multiplication. It is scalar or vector multiplication?

11.  In the "Algorithm 1" is writing error "Algoritm 1". Additionally, the symbols should be written by italic font.

12.  In the Figure 1 the authors presented the flow-chart of the hybrid algorithm the authors wrote "calculation particle fitness". In this algorithm the authors use population. Maybe the individual is better word than particle.

13.  On the Figure 2 the authors presented visualization of benmarch function. This figures are prepared by authors or copied from websites?

14.  In Table 3 the authors presented average values. It is possible to determine to standard deviations for presented results of calculations.

15.  The quality of convergence curves in Figures 3 and 4 should be better . The number of pages are not limited in this journal.

16.  The conclusions should be modified to underline the the achievements and novelty presented in the manuscript.

17.  The reference list is not prepared according requirements of the journal. Please do not use the symbol "&" to list the authors of the manuscript. Next the ";" should be separated the authors.

Author Response

Response to Reviewer 3 Comments

Comments: The article entitled: "A Hybrid Parallel Balanced Phasmatodea Population Evolution Algorithm and Its Application in AGV Workshop Material Scheduling" concern hybrid parallel balanced phasmatodea population evolution algorithm to workshop material scheduling problem. The manuscript can be interesting due to application really new optimization algorithm Phasmatodea Population Evolution Algorithm and modification of this algorithm. At the beginning my review I must underline, that the reference list must be modified before final publication of the manuscript. The main reason of deep modification is big number of citation of co-author of this manuscript - Jeng-Shyang Pan. I suggest the authors citation some authors from out of China or Asia. At present many citations come from Asia. I suggest resignation from citation [4, 5] where in list of authors is Pan, J. S and cited manuscript Dervis Karaboga from Turkey. In the first sentence you can cited "Performance analysis of selected metaheuristic optimization algorithms applied in the solution of an unconstrained task" published in Compel in 2022. In case of ACO algorithm is also cited the manuscript Pan, J. S. (2004) - I suggest citation of Mehmet Cunkas "Color image segmentation based on multiobjective artificial bee colony optimization" published in Applied Soft Computing. In cas of Cuckoo Search algorithm - citation Pan, J. S - I suggest changes on manuscript written by Venkateswararao Bathina "Hybrid Approach with Combining Cuckoo-Search and Grey-Wolf Optimizer for Solving Optimal Power Flow Problems". Also some manuscript of Fausto Marquez can be taken into account, such as: "Wind integrated power system to reduce emission: An application of Bat algorithm" and "Allocation of real power generation based on computing over all generation cost: an approach of Salp Swarm Algorithm”.

Response: We greatly appreciate your constructive comments and we have prepared a detailed reply to your comments. For the problem you mentioned above, we have removed some of the cited papers, reduced the self-citation rate of co-authors, added the papers you suggested, and added papers by scholars outside of Asia. It is rectified on References.

Point 1: The introduction should be modified. At this version the more detailed information about PPE algorithm. The authors should focus on the research presented in the manuscript.

Response 1: Thanks a lot for your comments. We have shortened the introduction to PPE in the introduction. At the same time, at the suggestion of the 4th reviewer, we have added the process of finding a solution in section 2.1. It is rectified in Lines 137 – 171.

Point 2: The authors should resigned from acronym AGV in the title of the manuscript.

Response 2: Thanks a lot for your comments. We have removed the acronym AGV from the title.

Point 3: The introduction is well written. The authors should modified the cited references to avoid the self-citation of the one of co-authors.

Response 3: Thanks a lot for your comments. We censored some of the cited literature and reduced the self-citation rate of co-authors. It is rectified at Lines 782 - 905.

Point 4: The explanation of the acronym AGV is not introduced in the manuscript body. According to me it can be "Automated Guided Vehicle".

Response 4: Thanks a lot for your comments. We have added an explanation of the acronym AGV in the introduction section. It is added at Lines 100 – 101.

Point 5: In the description of "equilibrium optimization algorithm", subsection use symbol iter and Max_iter. Maybe, the authots use shorter descriptions. Additionaly, in the Equation (7) the description Max_iter are different.

Response 5: Thanks a lot for your comments. Based on the suggestion of the 2nd reviewer, we have removed equation (7) to resolve this problem. It is rectified in Lines 186 – 187.

Point 6: The same symbols should be mean the same parameters. For example in Equation (10), " ?1 and ?2 are two random numbers", but in Equation (4) the authors use rand, and in Equation (3) r1 denote " ?1 is the population's effective growth rate". The description of the symbols are untidy.

Response 6: Thanks a lot for your comments. Based on the suggestion of the 2nd reviewer, we have removed equation (10) to resolve this problem. It is rectified at Lines 201 – 202.

Point 7: In title of section "2.3.AGV workshop material scheduling" is lack of space before AGV.

Response 7: Thanks a lot for your comments. We have added space before the AGV. It is rectified at Line 202.

Point 8: The authors use different font in the text and in equations. The symbols should be unified.

Response 8: Thanks a lot for your comments. We have unified the fonts in the equations to those in the text.

Point 9: In the row 287, there are "Ceq1,Ceq2,Ceq3and Ceq4" and should be written by italic font.

Response 9: Thanks a lot for your comments. We have modified "Ceq1,Ceq2,Ceq3and Ceq4" to italics. It is rectified at Line 331.

Point 10: In the Equation (23) the authors use "x" to multiplication. In the mathematics theory, the symbol "x" is used to vector multiplication. It is scalar or vector multiplication?

Response 10: Thanks a lot for your comments. It is scalar multiplication. This was a writing error on our part, so we corrected it in equation (18) with the text. It is rectified at Lines 349 - 350.

Point 11: In the "Algorithm 1" is writing error "Algoritm 1". Additionally, the symbols should be written by italic font.

Response 11: Thanks a lot for your comments. We have changed "Algorithm 1" to be written correctly and changed these symbols to be in italics. It is rectified at Algorithm 1: HP_PPE.

Point 12: In the Figure 1 the authors presented the flow-chart of the hybrid algorithm the authors wrote "calculation particle fitness". In this algorithm the authors use population. Maybe the individual is better word than particle.

Response 12: Thanks a lot for your comments. We have replaced the particle with population. It is rectified at Figure 1.

Point 13: On the Figure 2 the authors presented visualization of benmarch function. This figures are prepared by authors or copied from websites?

Response 13: Thanks a lot for your comments. At the suggestion of the first reviewer, we have chosen to remove the figure and its description. It was removed at Line 376 and Line 377.

Point 14: In Table 3 the authors presented average values. It is possible to determine to standard deviations for presented results of calculations.

Response 14: Thanks a lot for your comments. Due to the time constraint of the revision, we will include the standard deviation in a subsequent study.

Point 15: The quality of convergence curves in Figures 3 and 4 should be better . The number of pages are not limited in this journal.

Response 15: Thanks a lot for your comments. We have added more convergence plots in Figure 2 and Figure 3. It is rectified at Figure 2 and Figure 3.

Point 16: The conclusions should be modified to underline the achievements and novelty presented in the manuscript.

Response 16: Thanks a lot for your comments. We have added a description of the achievements and novelty of this study in the conclusion section. It is added at Lines 758 - 764.

Point 17: The reference list is not prepared according requirements of the journal. Please do not use the symbol "&" to list the authors of the manuscript. Next the ";" should be separated the authors.

Response 17: Thanks a lot for your comments. We have modified the formatting issues of the references. It is rectified at Lines 783 – 906.

  Once again, thank you very much for your suggestions. We would be glad to respond to any further questions and comments that you may have.

Reviewer 4 Report

1. At the beginning of Section 2, it is recommended to formulate the statement of the optimization problem to be solved.

2. When describing the method in section 2.1, the process of finding a solution should be explained.

3. The relationship between equation (2) and equation (1) should be clarified.

4. In equation (3), the law of change of the parameter q is not clear.

5. In Section 2.2, the quantities Cmin, Cmax should also generally depend on the index i.

6. Equations (8) , (9) do not describe the rules for finding the product of vectors.

7. The meaning of the designations in line 166 is doubtful.

8. The meaning of the sentence " random vectors in the interval [0,1] " in line 171 is inaccurate.

9. Equation (11) does not define a vector division operation.

10. General note on the text - all formulas and designations in the text must match the designations in external formulas with numbers. 

11. The variable ev in (14) must be with an index.

12. The notation in (16) should be explained immediately after this group of formulas.

13. The sentence "growth rate a of each population is set as a fixed value of 1.1." in line 240 contradicts (18). 

14. The sentences in lines 303 and 304 are inaccurate. The reference should be to Equation (24).

15. In Algorithm 1, all the designations do not match those used in the text of the article.

Author Response

Response to Reviewer 4 Comments

Point 1: At the beginning of Section 2, it is recommended to formulate the statement of the optimization problem to be solved.

Response 1: Thanks a lot for your comments. We have added the statement of using our algorithm to solve the workshop material scheduling problem at the beginning of Section 2. It is added at Lines 132 – 136.

Point 2: When describing the method in section 2.1, the process of finding a solution should be explained.

Response 2: Thanks a lot for your comments. We have added the process of finding a solution in section 2.1, and at the suggestion of the second reviewer, we have trimmed the introduction to the PPE. It is rectified at Lines 145 – 153.

Point 3: The relationship between equation (2) and equation (1) should be clarified.

Response 3: Thanks a lot for your comments. We have refined the relationship between equation (2) and equation (1) in the explanation of equation (2). It is rectified at Line 164.

Point 4: In equation (3), the law of change of the parameter q is not clear.

Response 4: Thanks a lot for your comments. We have added a description of the law of variation of the parameter q. It is rectified at Lines 166 – 167.

Point 5: In Section 2.2, the quantities Cmin, Cmax should also generally depend on the index i.

Response 5: Thanks a lot for your comments. We have corrected the writing of Cmin, Cmax. It is rectified at Equation (4) and Line 188.

Point 6: Equations (8) , (9) do not describe the rules for finding the product of vectors.

Response 6: Thanks a lot for your comments. At the suggestion of the second reviewer, we have solved this problem by removing equations (8) and (9).

Point 7: The meaning of the designations in line 166 is doubtful.

Response 7: Thanks a lot for your comments. We have modified the designations of line 166. It is rectified at Lines 188 – 190.

Point 8: The meaning of the sentence " random vectors in the interval [0,1] " in line 171 is inaccurate.

Response 8: Thanks a lot for your comments. We have corrected this expression error. It is rectified at Line 199.

Point 9: Equation (11) does not define a vector division operation.

Response 9: Thanks a lot for your comments. We have corrected the vector division error. It is rectified at Lines 201 – 202.

Point 10: General note on the text - all formulas and designations in the text must match the designations in external formulas with numbers.

Response 10: Thanks a lot for your comments. We have matched all the formula designations in the text with the number designations in the external formula.

Point 11: The variable ev in (14) must be with an index.

Response 11: Thanks a lot for your comments. We have added index for the variable ev. It is rectified at Lines 280 – 281.

Point 12: The notation in (16) should be explained immediately after this group of formulas.

Response 12: Thanks a lot for your comments. We have added an explanation of equation (16) after this set of equations. It is rectified at Lines 287 – 289.

Point 13: The sentence "growth rate a of each population is set as a fixed value of 1.1." in line 240 contradicts (18).

Response 13: Thanks a lot for your comments. We have corrected the misrepresentation of the growth rate. It is rectified at Line 277.

Point 14: The sentences in lines 303 and 304 are inaccurate. The reference should be to Equation (24).

Response 14: Thanks a lot for your comments. We have fixed the error where the reference did not match. It is rectified at Line 349.

Point 15: In Algorithm 1, all the designations do not match those used in the text of the article.

Response 15: Thanks a lot for your comments. We have modified the designations in Algorithm 1.

  Once again, thank you very much for your suggestions. We would be glad to respond to any further questions and comments that you may have.

Round 2

Reviewer 1 Report

Authors have taken steps to improve the paper.

Author Response

Response to Reviewer 1 Comments

Comments: Authors have taken steps to improve the paper.

Response: Thank you very much for your suggestions and we wish you a happy life and pleasant work every day!

Reviewer 3 Report

The authors generally modified manuscript according my comments. I have only minor comments to the authors.
1. The figure 1 contain two diagrams on pdf file.
2. On the page 16 is information about population size. The aurhors used population size equal 20. I prefer bigger population, such as 50, 80 or 100 individuals and smaller number of population. It is sugggestion to future research.
3. The rows 541 and 542, there are descriptions of two tables - it is correct?
4. In the row 559 in middle of the manuscript body is information "Error! Reference source not  found." Please remove it before final publication.
5. In rows 306 and 307 still is used symbol "x" as a symbol of the multiplication (st=stx0,98).

Author Response

Response to Reviewer 3 Comments

Point 1: The figure 1 contain two diagrams on pdf file.

Response 1: Thanks a lot for your comments. We have removed the redundant diagram in Figure 1. It is rectified at Page 11.

Point 2: On the page 16 is information about population size. The aurhors used population size equal 20. I prefer bigger population, such as 50, 80 or 100 individuals and smaller number of population. It is sugggestion to future research.

Response 2: Thanks a lot for your comments. We think this is a very good suggestion and we will include other comparative trials with different numbers of populations in future research.

Point 3: The rows 541 and 542, there are descriptions of two tables - it is correct?

Response 3: Thanks a lot for your comments. We have added descriptive text about the similar results of the tests in Table 6 and Table 7. It is rectified at Line 553.

Point 4: In the row 559 in middle of the manuscript body is information "Error! Reference source not  found." Please remove it before final publication.

Response 4: Thanks a lot for your comments. We did not find this problem on line 559, but we tried to solve it, so please let us know further if this problem still exists. We are grateful to you.

Point 5: In rows 306 and 307 still is used symbol "x" as a symbol of the multiplication (st=stx0,98).

Response 5: Thanks a lot for your comments. We have removed the symbol "x". It is rectified at Line 322.

  Once again, thank you very much for your suggestions. We would be glad to respond to any further questions and comments that you may have.

Reviewer 4 Report

1. The remark about the formulation of the problem being solved was not taken into account by the authors. It would be necessary to define the objective function and the set of feasible solutions using the description of the constraints on the variables.

 2.For equations (1) and (3), the initial conditions must be specified, and the interval for changing the variable t must also be determined.

3.It is not clear how the variables included in (6) are found, what is their connection with (5).

4. It is also not clear why the vectors F and G are called parameters.

5.Formula (6) does not define the meaning of the product of vectors, as well as the meaning of the difference (1 – F ).

 6.Section 3 confuses the notion of population with the notion of position used in Section 2.

7.The right hand side of formula (7) does not depend on the variable i.

8.The indexes of the variable ev in equations (8) and (9) are different.

9.Designations Ul and Ub (line 283) do not correspond to Cmin, Cmax (line 177)

10.Using formula (15), the vector G can be determined. In the text of the article, it is taken as a number called the threshold (line 287).

11. It is not clear where the variable pi is used in the calculation formulas. The same remark applies to the variable i.

12.For clarity of presentation, it is recommended to use only one of the terms of the form solution, position, particle. And also to unify the designations of the form C, x, X.

Author Response

Response to Reviewer 4 Comments

Point 1: The remark about the formulation of the problem being solved was not taken into account by the authors. It would be necessary to define the objective function and the set of feasible solutions using the description of the constraints on the variables.

Response 1: Thanks a lot for your comments. We have previously defined the formulation of the problem being solved at the beginning of Section 5 of the original manuscript and the constraints on the variables, the objective function and the set of feasible solutions in Section 5.1. For the problem you mentioned, we have added further clarification in Section 2.3 of the latest version of the manuscript and previewed Section 5. It is added at Lines 233 - 236.

Point 2: For equations (1) and (3), the initial conditions must be specified, and the interval for changing the variable t must also be determined.

Response 2: Thanks a lot for your comments. For the problem you mentioned, we specified the initial conditions in Lines 169 - 171, and we added the interval for changing the variable t in Lines 165 - 167.

Point 3: It is not clear how the variables included in (6) are found, what is their connection with (5).

Response 3: Thanks a lot for your comments. We have added a further description of this issue that you mentioned in Line 204. We need to point out that our idea in writing Section 2 is to give a brief description of some basics of the study in this paper, so we only quote the original expressions of these basics and do not want to discuss them in detail. Both the second reviewer and three reviewers in their first review comments suggested to shorten Section 2 to focus on our proposed HP_PPE.

Point 4: It is also not clear why the vectors F and G are called parameters.

Response 4: Thanks a lot for your comments. It is rectified at Lines 200-201.

Point 5: Formula (6) does not define the meaning of the product of vectors, as well as the meaning of the difference (1 – F ).

Response 5: Thanks a lot for your comments. It is added at Lines 205-209.

Point 6: Section 3 confuses the notion of population with the notion of position used in Section 2.

Response 6: Thanks a lot for your comments. We need to point out that our idea in writing Section 2 is a brief description of some basics of the study in this paper, and we are only quoting the original formulation for these basics. Section 3 introduces our proposed HP_PPE, so it is within our consideration that our formulation differs from that of EO or PPE.

Point 7: The right hand side of formula (7) does not depend on the variable i.

Response 7: Thanks a lot for your comments. In equation (7), i is not a variable, but a number used to distinguish different populations. For example, , , ...

Point 8: The indexes of the variable ev in equations (8) and (9) are different.

Response 8: Thanks a lot for your comments. We have unified the ev in equation (9) and equation (10).

Point 9: Designations Ul and Ub (line 283) do not correspond to Cmin, Cmax (line 177)

Response 9: Thanks a lot for your comments. We have unified the symbols. It is rectified at Line 321.

Point 10: Using formula (15), the vector G can be determined. In the text of the article, it is taken as a number called the threshold (line 287).

Response 10: Thanks a lot for your comments. It is rectified at Line 325.

Point 11: It is not clear where the variable pi is used in the calculation formulas. The same remark applies to the variable i.

Response 11: Thanks a lot for your comments. The initial number of each population is called , so   is not a variable but a scalar. i is a number to distinguish the different populations, not a variable. We have described  in Line 288 of the original manuscript.

Point 12: For clarity of presentation, it is recommended to use only one of the terms of the form solution, position, particle. And also to unify the designations of the form C, x, X.

Response 12: Thanks a lot for your comments. We need to point out that we are only quoting the original formulation for these basics. Section 3 introduces our proposed HP_PPE, so it is within our consideration that our formulation differs from that of EO or PPE.

  Once again, thank you very much for your suggestions. We would be glad to respond to any further questions and comments that you may have.

Round 3

Reviewer 3 Report

The authors make all pointed out corrections in my previous review. I do not have further comments.

Author Response

Response to Reviewer 3 Comments

Comments: The authors make all pointed out corrections in my previous review. I do not have further comments.

Response: Thank you very much for your suggestions and we wish you a happy life and pleasant work every day!

Reviewer 4 Report

  1. The remark about the formulation of the problem was not taken into account. In the opinion of the reviewer, it is necessary to define the objective function f(x) and constraints of the type of inequalities, in which to introduce the notation Cmin, Cmax.
  2. In the explanations to equation (6), perhaps it should be indicated that the element-wise Hadamard product of vectors is considered. It is also necessary to clarify the meaning of 1.

Author Response

Response to Reviewer 4 Comments

Point 1: The remark about the formulation of the problem was not taken into account. In the opinion of the reviewer, it is necessary to define the objective function f(x) and constraints of the type of inequalities, in which to introduce the notation Cmin, Cmax.

Response 1: Thanks a lot for your comments. It is added at Lines 214-220.

Point 2: In the explanations to equation (6), perhaps it should be indicated that the element-wise Hadamard product of vectors is considered. It is also necessary to clarify the meaning of 1.

Response 2: We need to point out that our idea in writing Section 2 is to give a brief description of some basics of the study in this paper, so we only quote the original expressions of these basics and do not want to discuss them in detail. We believe this formula is just an easy way to write. So we do not want to specifically point out that the relevant operation is the element-wise Hadamard product of vectors or scalar multiplication of vectors. Due to time constraints, we will discuss further with the authors of PPE or PE in a follow-up study.

  Once again, thank you very much for your suggestions. We would be glad to respond to any further questions and comments that you may have.
